# A time-resolved interaction analysis of Bem1 reconstructs the flow of Cdc42 during polar growth

Sören Grinhagens[1],*, Alexander Dünkler[1],*, Yehui Wu[1],*, Lucia Rieger[1],*, Philipp Brenner[1], Thomas Gronemeyer[1], Medhanie A Mulaw[2], Nils Johnsson[1]

**Cdc42 organizes cellular polarity and directs the formation of cellular structures in many organisms. By locating Cdc24, the source of active Cdc42, to the growing front of the yeast cell, the scaffold protein Bem1, is instrumental in shaping the cellular gradient of Cdc42. This gradient instructs bud formation, bud growth, or cytokinesis through the actions of a diverse set of effector proteins. To address how Bem1 participates in these transformations, we systematically tracked its protein interactions during one cell cycle to define the ensemble of Bem1 interaction states for each cell cycle stage. Mutants of Bem1 that interact with only a discrete subset of the interaction partners allowed to assign specific functions to different interaction states and identified the determinants for their cellular distributions. The analysis characterizes Bem1 as a cell cycle–specific shuttle that distributes active Cdc42 from its source to its effectors. It further suggests that Bem1 might convert the PAKs Cla4 and Ste20 into their active conformations.**

## Introduction

Bud formation, growth, and cell separation are the visible consequences of polar cell growth in the budding yeast (Bi & Park, 2012; Howell & Lew, 2012). Interactions between the involved cell polarity proteins might act as switches to drive these morphological alterations. Accordingly, changes in the composition and structure of the protein interaction network should correlate with the different phases of cell growth.

Yeast cells initiate bud formation at a predetermined site, expand the bud preferentially at its tip, switch in large buds to an isotropic growth, and finally reorient the growth axis during mitosis and cell separation (Howell & Lew, 2012). The Rho-like GTPase Cdc42 influences local cell expansion in all cell cycle phases by binding in its active, GTP-bound state to different effector proteins (Chiou

et al, 2017). $Cdc42_{GTP}$ instructs the organization of the septin- and actin cytoskeleton, the spatial organization of exocytosis, mating, osmolarity sensing, and mitotic exit (Pruyne et al, 2004; Bi & Park, 2012). Cdc24, the guanine-nucleotide-exchange factor (GEF) for Cdc42, and a variety of (GAPs) GTPase-activating protein adjust the concentration of $Cdc42_{GTP}$ at the cortex (Smith et al, 2002). The concentration of $Cdc42_{GTP}$ changes over the cell cycle and peaks at the G1/S and at anaphase (Atkins et al, 2013).

Bem1 is the central scaffold for proteins that organize polarized growth in yeast (Chenevert et al, 1992; Peterson et al, 1994; Bender et al, 1996; Matsui et al, 1996). Bem1 binds Cdc24, $Cdc42_{GTP}$, and several $Cdc42_{GTP}$ effector proteins (Bose et al, 2001; Irazoqui et al, 2003). The protein is part of the polarity cap during bud growth, cell separation, cell mating, and fusion and assists Cdc42 in the pheromone response-, the filamentous growth-, and the high osmolarity MAPK pathways (Lyons et al, 1996; Leberer et al, 1997; Winters & Pryciak, 2005; Tanaka et al, 2014).

During G1, Bem1 plays a key role in polarity establishment by forming a stable zone of $Cdc42_{GTP}$ at the cell cortex. Physically connecting Ccd24 to $Cdc42_{GTP}$, Bem1 organizes a positive feedback where $Cdc42_{GTP}$ attracts further Cdc24 to activate even more Cdc42 (Irazoqui et al, 2003; Kozubowski et al, 2008; Woods et al, 2015; Witte et al, 2017).

Bem1 consists of two N-terminally located (SH3) SRC homology 3 domains ($SH3_a$ and $SH3_b$), a lipid-binding (PX) phox homology domain, and a C-terminal (PB1) Phox and Bem1 domain ($PB1_{Bem1}$) (Bender et al, 1996; Matsui et al, 1996). $SH3_b$ interacts with well-characterized PxxP motifs in the p21 activated kinase (PAKs) Cla4 and Ste20, and the polarity proteins Boi1 and Boi2 (Bender et al, 1996; Bose et al, 2001; Winters & Pryciak, 2005; Gorelik & Davidson, 2012). $SH3_b$ harbors a C-terminal extension (CI) that binds $Cdc42_{GTP}$ (Yamaguchi et al, 2007; Takaku et al, 2010). $PB1_{Bem1}$ binds the C-terminal PB1 domain of Cdc24 with high affinity and localizes Cdc24 to sites of polar growth during all cell cycle stages (Butty et al, 2002; Woods et al, 2015; Witte et al, 2017). The mechanisms of Bem1's precisely regulated cellular distribution are, however, not fully understood (Woods et al, 2015; Meca et al, 2019).

Linking Cdc42 to Cdc24 might not suffice to explain the many functions of Bem1 during the other phases of the cell cycle (Atkins

[1]Department of Biology, Institute of Molecular Genetics and Cell Biology, Ulm University, Ulm, Germany   [2]Comprehensive Cancer Center Ulm, Institute of Experimental Cancer Research, Ulm University, Ulm, Germany

Correspondence: nils.johnsson@uni-ulm.de
*Sören Grinhagens, Alexander Dünkler, Yehui Wu, and Lucia Rieger contributed equally to this work

et al, 2008; Kozubowski et al, 2008; Li & Wedlich-Soldner, 2009). Instead, Bem1 was also shown to modestly stimulate Cdc24's GEF activity (Smith et al, 2013; Rapali et al, 2017). By simultaneously binding to Cla4/Ste20, active Cdc42, and Cdc24, Bem1 might also induce a negative feedback to tone down the activity of Cdc24 during later stages of the cell cycle (Gulli et al, 2000; Kozubowski et al, 2008; Kuo et al, 2014; Rapali et al, 2017).

Here, we probe the interaction network of Bem1 throughout polar growth and cytokinesis to correlate changes in composition and architecture of the network with changes in cellular morphology and the activities of its binding partners.

# Results

## A protein interaction map of Bem1

We searched for binding partners of Bem1 by performing a systematic split-ubiquitin (Split-Ub) screen of Bem1-$C_{ub}$-RUra3 (Bem1CRU) against 548 $N_{ub}$ fusion proteins known or suspected to be involved in different aspects of polarized growth in yeast (see the Materials and Methods section) (Johnsson & Varshavsky, 1994; Hruby et al, 2011). The screen identified besides known binding partners, Bud6, Msb1, Ras1, Ras2, Rga2, Nba1, Spa2, Cdc11, Fks1, and Bem1 as novel interaction partners of Bem1 (Fig 1A). We repeated the screen with mutants of Bem1 that either carried the well-characterized W192K exchange in $SH3_b$ (Bem1$_{WK}$CRU) or lacked the C-terminal PB1 domain, and thus, the binding site to Cdc24 (Bem1$_{\Delta PB1}$CRU) (Fig 1A and B). The comparison with Bem1CRU fusion revealed that Bem1$_{WK}$ lost its interactions to Boi1, Ste20, Cla4, Bud6, Nba1, and Bem1 and showed a strongly reduced binding to Boi2 but retained its interactions with Exo70, Cdc24, Cdc42, Cdc11, Rga2, Msb1, Ras2, and Ras1 (Fig 1A and B). Deleting the PB1 domain in Bem1$_{PB1\Delta}$CRU removed or strongly reduced the interactions of Bem1 to $N_{ub}$-Cdc24, -Cdc11, -Rga2, -Msb1, -Ras1, -Ras2, and -Exo70 (Fig 1A and B). Neither mutation visibly affected the interaction of Bem1 with Fks1, Cdc42, or Spa2 (Fig 1A).

## Dissection of the Bem1 interaction network

The Split-Ub assay detects direct and indirect protein interactions (Hruby et al, 2011; Johnsson, 2014). $SH3_b$ mediates the interactions between Bem1 and Boi1, Boi2 (Boi1/2), Nba1, or Bud6 (Fig 1A and B). Boi1/2 bind $SH3_b$ and interact directly with Bud6 and Nba1 (Bender et al, 1996; Kustermann et al, 2017). To test whether Boi1/2 link Bem1 to Nba1 or Bud6, we introduced Bem1CRU together with $N_{ub}$-Bud6, or $N_{ub}$-Nba1 in a strain that carried either a deletion of *BOI1* or *BOI2*, or a deletion of *BOI2* and the mutated binding site of Boi1 for Bem1 (*boi2Δ boi1$_{PxxP\Delta}$*). The Split-Ub assays confirmed that the interactions between Bem1 and Bud6 or Bem1 and Nba1 clearly depend on Boi1/2 (Fig 2A). The nearly complete loss of interaction between Bem1CRU and $N_{ub}$-Bem1 in a *boi2Δ boi1$_{PxxP\Delta}$* strain suggests that the Split-Ub detected Bem1–Bem1 interaction is predominantly mediated by the multimerization of the Boi proteins (Figs 1A and B and 2A) (Kustermann et al, 2017). In contrast to the Nba1-Boi1/2-Bem1 or the Bem1-Boi1/2–Bem1 complex, the interaction between Bud6 and Bem1 was already lost upon deleting either *BOI1* or *BOI2* (Fig 2A).

The Bni1–Bud6 complex nucleates the polymerization of linear actin filaments (Graziano et al, 2011, 2013). Full activity of Bni1 requires its association with a Rho-GTPase (Evangelista et al, 1997; Dong et al, 2003). Bud6 consists of a C-terminal actin- and Bni1-binding domain and an N-terminal region of unknown function (Tu et al, 2012). Testing $N_{ub}$-Boi1 and $N_{ub}$-Boi2 against CRU fusions to the N- and C-terminal fragments of Bud6 located the binding sites for Boi1/2 to its N-terminal 364 residues (Fig 2B). The GST fusion to this fragment precipitated Boi1- and Boi2-GFP from yeast extracts, thus providing an independent confirmation of the Split-Ub analysis and for the existence of a novel potential actin nucleation complex (Fig 2C and D) (Glomb et al, 2020).

Nba1 was reported to down-regulate the concentration of active Cdc42 during cytokinesis (Meitinger et al, 2014). Testing N- and C-terminal fragments of $N_{ub}$-Boi1/2 located the binding sites for Nba1 to the SH3 domains of both proteins (Fig 3A). Introducing single residue exchanges into the SH3 domains of $N_{ub}$-Boi1/2 (Boi1$_{WK}$, Boi1$_{PL}$, Boi2$_{WK}$, and Boi2$_{PL}$) also abolished the interactions with Nba1CRU (Fig 3B) (Larson & Davidson, 2000). Testing a C-terminal fragment of Nba1 as $N_{ub}$ fusion against Boi1- and Boi2CRU restricted the binding motif for both SH3 domains between residues 256 and 501 of Nba1 (Fig 3B and C). This region harbors a consensus-binding motif for the SH3 domains of Boi1/2 (Tonikian et al, 2009). Removing this PxxP site (Nba1$_{PxxP\Delta}$, Fig 3C) impaired the interaction between the corresponding $N_{ub}$-Nba1$_{PxxP\Delta}$ and the $C_{ub}$ fusions of Boi1/2 or Bem1 (Fig 3B and C). Surface plasmon resonance spectrometry determined the $K_D$s between the PxxP site (6xHIS-Nba1$_{202-289}$SNAP) and SH3$_{Boi1}$ and SH3$_{Boi2}$ to ~0.74 $\mu M$ (n = 3) and 1.97 $\mu M$ (n = 2), respectively (Fig 3D).

Nba1 is attached to the bud neck through a direct interaction with Gps1 (Meitinger et al, 2014). Split-Ub analysis reproduced the interactions of Gps1CRU with the $N_{ub}$-fusions of Nba1, and Nba1$_{PxxP\Delta}$, and revealed novel interactions between Gps1CRU and $N_{ub}$-Boi1/2 (Fig 3B). Mutations in the SH3 domains of $N_{ub}$-Boi1/2 abolished their interactions with Gps1CRU (Fig 3B). A C-terminal fragment of Gps1 still interacted as CRU fusion with $N_{ub}$-Nba1 and $N_{ub}$-Boi1 (Fig 3E). The interaction between Gps1$_{537-758}$CRU and $N_{ub}$-Boi1 was lost in an *nba1Δ*-strain. The experiments support the existence of a protein complex connecting Bem1 with Gps1 through Boi1/2 and Nba1 (Fig 3E and F).

The interaction signal between Bem1CRU and $N_{ub}$-Cdc11 was lost upon removal of PB1$_{Bem1}$ (Fig 1). Cdc11 interacts directly with Cdc24 (Chollet et al, 2020). We conclude that the interaction between Cdc11 and Bem1 occurs most likely through Cdc24 in a Bem1-Cdc24-Cdc11 complex.

## Functional dissection of Bem1 interaction states

Bem1 is thought to coordinate the activities of its ligands by bringing them into close spatial proximity. To define which combination of binding sites and partners constitute the essential configurations of the Bem1 complex, we first tested mutants and fragments for their ability to complement a deletion of *BEM1*. *BEM1* is not essential in all yeast strains but is required for cell survival in the strain JD47 (Fig 4A) (Dowell et al, 2010). *bem1Δ* cells can be rescued by a deletion of the Cdc42 GAP Bem3 but not by the deletion of the Cdc42 GAP Bem2 (Fig 4A) (Laan et al, 2015). Mutating

# A

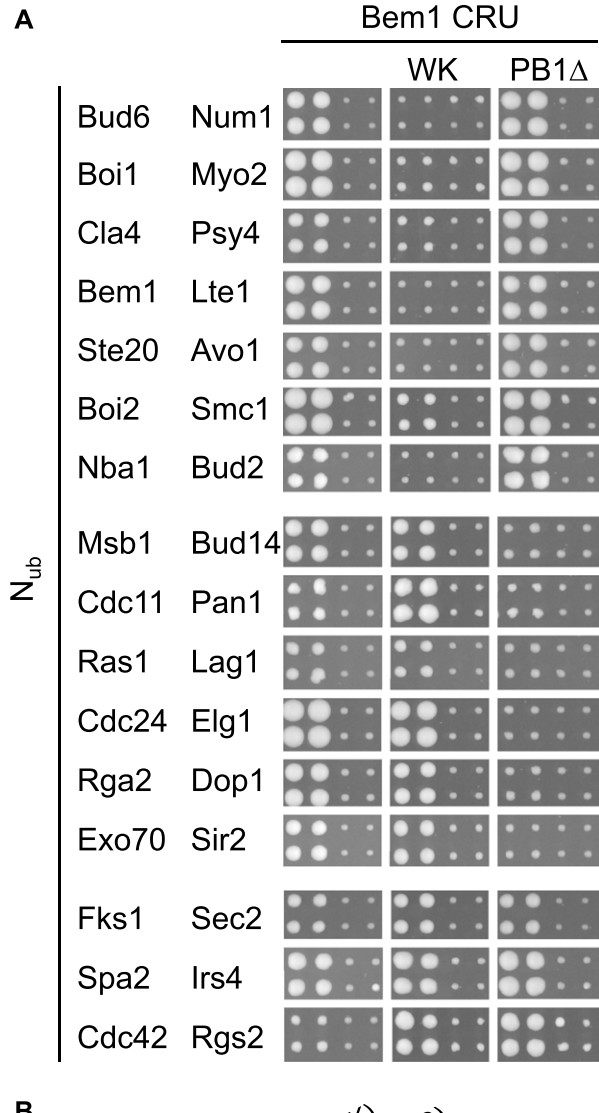

# B

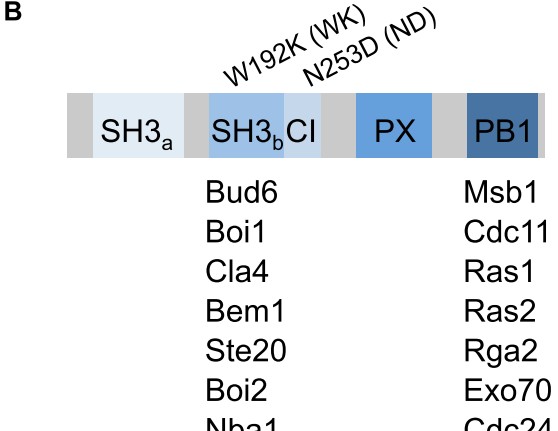

**Figure 1.  Interaction partners of Bem1.**
**(A)** Yeast cells carrying Bem1CRU or either of its two mutants Bem1WK- and Bem1PB1ΔCRU were independently mated four times with Nub fusion expressing strains. Interaction is indicated by growth of the four matings on SD medium containing 5-fluoro-orotic acid. Shown are the cut outs of the quadruplets expressing the Nub fusion of the interacting protein on the left, next to a fusion that does not interact. **(B)** The domains of Bem1 and the positions of the residue exchanges of the *bem1*-alleles used in this work. The domain-specific interaction partners of (A) are listed below the respective domains.

Bem3's GAP domain or impairing its interaction with lipids was sufficient to restore viability to *bem1*Δ cells (Fig S1A). We conclude that a high concentration of cortical Cdc42GTP can compensate for the loss of Bem1.

The CRIB domain of Gic2 (Gic2PBD) interacts with Cdc42GTP and has been tagged with red fluorescent protein to monitor active Cdc42 in living yeast cells (Brown et al, 1997; Orlando et al, 2008; Atkins et al, 2013; Okada et al, 2013). The overexpression of Gic2PBD is toxic in certain yeast mutants and can be compensated by increasing the amount of Cdc42 (Brown et al, 1997). We introduced Gic2PBD under control of the methionine-sensitive $P_{MET17}$- promoter in *bem1*Δ *bem3*Δ cells. Omitting methionine in the media increased the expression of Gic2PBD and eliminated the positive effect of the *BEM3* deletion on the survival of *bem1*Δ cells (Figs 4A and S4A). The results imply that Gic2PBD might reduce the free pool of Cdc42GTP at the cell cortex. It follows that Bem1 is needed to stimulate the synthesis and/or to improve the effective use of this pool.

A fragment of Bem1 (Bem1145-551) that covers SH3bCI, the PX-, and the PB1 domain and thus keeps the majority of all detected interactions, complemented *bem1*Δ cells (Figs 4B and S2). This region could be further divided into two independently complementing fragments: the SH3bCI domain (Bem1145-268) with its binding sites for Cdc42 and for its PxxP ligands Ste20, Cla4, Boi1/2, and the C-terminal fragment containing the PX- and the PB1 domain (Bem1268-551) with its binding sites for Cdc24, lipids, and the other PB1-domain ligands (Figs 1A, 4B and C, and S2). Expressing both fragments together complemented *bem1*Δ cells much better than each fragment alone (Fig 4C).

The autonomy of the central SH3bCI-domain was unexpected. Single mutations that interrupt the binding of SH3bCI either to the PxxP ligands (SH3bWKCI) or to Cdc42GTP (SH3bCIND) interfered with the fragment's ability to rescue *bem1*Δ cells (Figs 4E and S2) (Yamaguchi et al, 2007; Gorelik & Davidson, 2012).

The existence of two independently complementing regions explains why none of the single mutations in the SH3bCI domain or the deletion of the PB1 domain eliminated the functionality of the otherwise full-length Bem1 (Fig 4B and E). By expressing increasing amounts of Gic2PBD, we tested the functionality of the *bem1WK*-, *bem1KA*-, *bem1ND*-, or *bem1WK ND* alleles under conditions of limiting Cdc42 (Fig 4D). All interaction-interfering mutations drastically decreased the tolerance of the cells toward Gic2PBD overexpression. The allele *bem1WK ND* bearing both mutations in the SH3bCI domain conferred a higher sensitivity than the singly mutated *bem1WK*- or *bem1ND* allele. Overexpressing Bem3 and thus reducing Cdc42GTP at the cortex by different means had a similar impact (Fig S1B).

We introduced the WK and ND mutations in the different *BEM1* copies of a diploid cell to test trans-complementation of the co-expressed Bem1WK and Bem1ND. The undiminished sensitivity of these cells toward Gic2PBD overexpression showed that complementation does not occur in trans and that both binding sites operate within the same Bem1 molecule (Fig 4D).

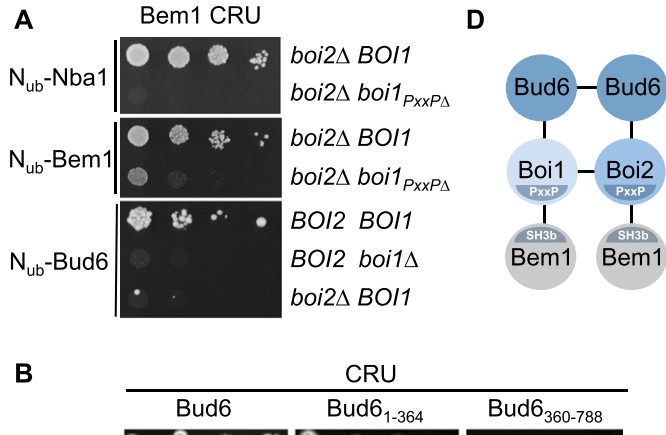

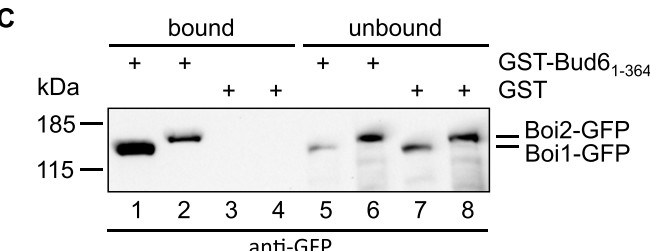

**Figure 2. Characterization of the Bem1-Bud6 interaction state.**
**(A)** Yeast cells carrying the indicated mutations were co-expressing CRU fusions to Bem1 together with the indicated $N_{ub}$ fusions. Cells were grown to an $OD_{600}$ of 1 and spotted in 10-fold serial dilutions onto medium containing 5-fluoro-orotic acid. Interactions are indicated by the growth of the yeast cells. **(B)** As in (A) but with yeast cells co-expressing CRU fusions to $Bud6_{1-364}$ or $Bud6_{360-788}$, together with $N_{ub}$ fusions to Boi1 and Boi2. **(C)** Extracts of yeast cells expressing either Boi1-GFP (lanes 1, 3, 5, and 7) or Boi2-GFP (lanes 2, 4, 6, and 8) were incubated with GST- (lanes 3, 4, 7, and 8) or GST-$Bud6_{1-364}$–immobilized (lanes 1, 2, 5, and 6) sepharose beads. Bound (lanes 1–4) and unbound (lanes 5–8) fractions were analyzed by anti-GFP antibodies after SDS–PAGE and transfer onto nitrocellulose. **(D)** Model of a potential regulator of actin nucleation. Bud6 is known to homodimerize, whereas Boi1 and Boi2 either homo- or heterodimerize. Bud6 binds and stimulates the yeast formin Bni1 (not shown).

## Functional annotation of the SH3$_b$ interactions

Our findings imply that delivering Cdc42$_{GTP}$ to one or more of its SH3$_b$ ligands constitutes the essential activity of Bem1. Which is the essential binding partner of SH3$_b$? The closely spaced SH3$_b$ and CI domains of Bem1 mirror the Cdc42$_{GTP}$ binding of the three of its four ligands Cla4, Ste20, and Boi1. The binding of Cdc42 to Boi2 was not yet investigated. None of the four SH3$_b$ ligands are essential. Cells, however, do not tolerate the loss of both PAKs, or of both Boi-proteins (Cvrckova et al, 1995; Bender et al, 1996). To identify the interactions whose loss could phenocopy the WK mutation in SH3$_b$, we introduced mutations in *CLA4* (*cla4$_{F15AAA/PPF451L}$* = *cla4$_{PPAFL}$*), *STE20* (*ste20$_{F470L\ P475T}$* = *ste20$_{FLPT}$*), and *BOI1* (*boi1$_{PxxP\Delta}$*) that specifically reduce their affinities to SH3$_b$ (Bender et al, 1996; Kozubowski et al, 2008; Gorelik & Davidson, 2012) (Fig S3). The mutations in *STE20* and *CLA4* did not impair their interaction with Nbp2, a further ligand of their PxxP motifs (Fig S3) (Winters & Pryciak, 2005; Hruby et al, 2011; Gorelik & Davidson, 2012). Which

of the SH3$_b$ interactions become essential under conditions of Gic2$_{PBD}$ overexpression? Gic2$_{PBD}$ overexpression killed cells lacking *CLA4* and the SH3$_b$-binding motif of Ste20, or cells lacking *STE20* and the SH3$_b$-binding motifs of Cla4, or cells co-expressing *cla4$_{PPAFL}$* with *ste20$_{FLPT}$* (Figs 5A and S4A). Cells lacking *BOI2* and the Bem1-binding site in Boi1 were not affected by Gic2$_{PBD}$ overexpression (Fig 5A). The pleckstrin homology (PH) domain of Boi1 (PH$_{Boi1}$) binds lipid and Cdc42$_{GTP}$ (Bender et al, 1996; Kustermann et al, 2017). Overexpression of PH$_{Boi1}$ killed cells lacking the physical connection of Bem1 to Ste20 and Cla4 but does not affect cells lacking the connection of Bem1 to Boi1/2 (Fig S1D). A simultaneous overexpression of Cdc42 suppressed the toxic effect of PH$_{Boi1}$ on *ste20$\Delta$ cla4$_{PPAAFL}$* cells (Fig S1E). We conclude that the PAKs are the essential ligands of the SH3$_b$ domain of Bem1 under conditions of limiting concentrations of active Cdc42.

Cells without Cla4 or its kinase activity do not correctly assemble septins and display elongated buds (Holly & Blumer, 1999; Weiss et al, 2000). These phenotypes were recapitulated in *bem1$_{WK}$$^-$* or in *bem1$_{ND}$* cells, or in *cla4$_{PPAAFL}$* cells upon overexpression of Gic2$_{PBD}$ (Fig 5B–D). In contrast, Gic2$_{PBD}$ overexpression did not affect cellular morphology or septin structure of *ste20$_{FLPT}$$^-$* or *boi2$\Delta$ boi1$_{PxxP\Delta}$* cells (Figs 5C and S4C). The experiments prove that the Cla4–Bem1–Cdc42$_{GTP}$ complex is important during incipient bud site- and septin-assembly.

A deletion of *STE20* rescues a strain that is arrested at cytokinesis by the loss of the cytokinesis factors Hof1 and Cyk3 (Atkins et al, 2013; Onishi et al, 2013). It is speculated that Cdc42 inhibits secondary septum formation through activation of Ste20. The loss of Ste20 might prematurely activate secondary septum formation, thus compensating the lack of primary septum. Fig 5E shows that *ste20$_{FLPT}$* also rescues *hof1$\Delta$ cyk3$\Delta$* cells. Accordingly, the Ste20–Bem1 complex is functionally relevant during cell separation. The same interaction is important for the fusion of cells during mating (Fig S4B) (Winters & Pryciak, 2005). Again, overexpression of Gic2$_{PBD}$ potentiated the effect of the *ste20$_{FLPT}$* allele (Fig S4B).

The Boi proteins stimulate the fusion of secretory vesicles with the plasma membrane (Kustermann et al, 2017; Masgrau et al, 2017). The small difference in the rate of bud length extension between *boi2$\Delta$* and *boi2$\Delta$ boi1$_{PxxP\Delta}$* cells suggests that the interactions between Boi1/2 and Bem1 only modestly affect this activity (Fig S4D).

### Boi1/2 anchor Bem1 at the bud tip

SH3$_b$CI-GFP is the minimal fragment that fully recapitulates the cellular distribution of Bem1-GFP (Fig 6A). The cortical targeting of SH3$_b$CI required the ligands of SH3$_b$ as SH3$_{bWK}$CI-GFP stayed cytosolic throughout the cell cycle (Fig 6A). SH3$_b$CI$_{ND}$-GFP was still concentrated at bud neck and tip. Under the assumption that the N253D exchange completely abrogates Cdc42$_{GTP}$ binding, we conclude that Cdc42 does not contribute to the cortical localization of SH3$_b$CI. To find out which of the four SH3$_b$ ligands influences the distribution of SH3$_b$CI-GFP, we expressed SH3$_b$CI-GFP in cells each lacking a specific SH3$_b$-binding site. The analysis identified Boi1/2 as the receptor for SH3$_b$CI at the cortex and bud neck (Fig 6B).

Bem1$_{WK}$-GFP was barely detected at the bud neck (Fig 6C, see also Fig 7), whereas its fluorescence signal at the tip of small and large buds was only modestly reduced (Fig 6C). To obtain a more quantitative measure of tip adherence, we compared the FRAPs

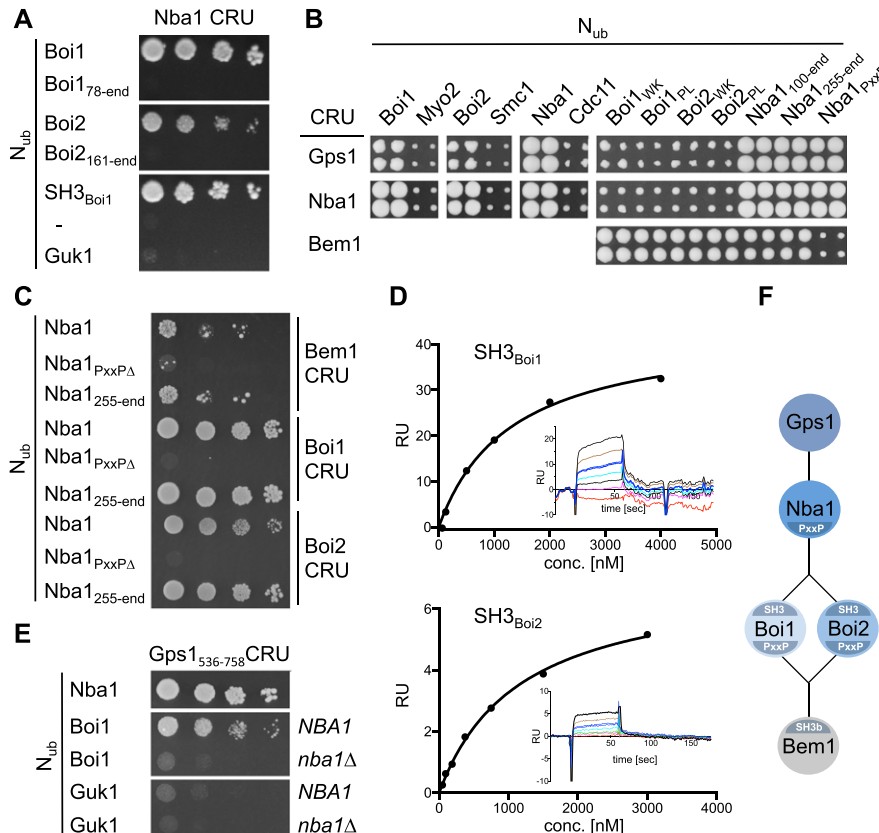

**Figure 3. Characterization of the Bem1-Nba1 interaction state.**
**(A)** Split-Ub assay as in Fig 2A, but with cells co-expressing CRU fusions to Nba1 together with the indicated $N_{ub}$ fusions. $N_{ub}$-Guk1: negative control. **(B)** a-yeast cells expressing the indicated CRU fusions were mated with α-yeast cells expressing the indicated $N_{ub}$ fusions and spotted on 5-fluoro-orotic acid medium as in Fig 1A. **(C)** As in (A) but with yeast cells co-expressing CRU fusions to Bem1, Boi1, or Boi2 together with $N_{ub}$ fusions to Nba1 or its mutants. **(D)** Surface plasmon resonance analysis of the interaction between 6xHIS-Nba1$_{202-289}$-SNAP and the chip-coated SH3 domains of Boi1 or Boi2. Shown are representative plots of the surface plasmon resonance signals as response units against the concentrations of 6xHIS-Nba1$_{202-289}$-SNAP. Corresponding sensograms are shown as insets. **(E)** As in (A) but with yeast cells containing or lacking *NBA1* and co-expressing Gps1$_{536-758}$CRU with the indicated $N_{ub}$ fusions. **(F)** Cartoon of the Nba1-Bem1 interaction state. The postulated indirect interaction between Bem1 and Gps1 was not experimentally observed but inferred from the Nba1-dependent interaction between Boi1/2 and Gps1 (see also Fig 7).

between the cortex-localized Bem1-GFP, Bem1$_{WK}$-GFP, and Bem1$_{ND}$-GFP (Fig 6D). The halftime of recovery was not changed by the N253D mutation in Bem1 ($t_{1/2}$ = 8.82 ± 1.12 s), whereas the W192K exchange reduced $t_{1/2}$ to 4.48 ± 0.46 s (Fig 6D). A similar reduction in $t_{1/2}$ was observed when the FRAPs of Bem1-GFP were compared between *boi2Δ* cells ($t_{1/2}$ = 7.35 ± 0.49 s) and *boi2Δ* cells lacking the Bem1-binding sites in Boi1 ($t_{1/2}$ = 5.42 ± 0.49 s) (Fig 6D). The FRAP of the cortex-localized Bem1 was not changed in cells where the interactions between Bem1 and the PAKs were eliminated (*ste20Δ cla4$_{PPAAFL}$*) (Fig 6D).

Boi1 and Boi2 associate with the bud cortex mainly through their Cdc42$_{GTP}$- and lipid-binding PH domains (Hallett et al, 2002). Bem1 also contains multiple phospholipid-binding sites (Hallett et al, 2002; Meca et al, 2019). Boi1/2 and Bem1 might, thus, cooperatively recruit each other to the bud cortex. In accordance, Boi1$_{PxxPΔ}$-GFP carrying a mutated Bem1-binding site was significantly more mobile than the native protein ($t_{1/2}$ = 16.43 ± 1.86 s versus $t_{1/2}$ = 10.58 ± 0.97 s) (Fig 6E). Boi1-GFP was also slightly less focused at the bud cortex of *bem1$_{WK}$* cells. *bem1$_{WK}$* did not detectably influence the cortical localization of Ste20-GFP or Cla4-GFP (Fig 6F and G) (Winters & Pryciak, 2005).

### Nba1 and Fir1 anchor Boi1/2-Bem1-Cdc24 at the bud neck

Bem1 leaves the cortex during mitosis and arrives at the bud neck shortly before the acto-myosin ring contraction is completed (Fig S4E). Boi1/2 link Bem1 and Cdc24 to the neck (Figs 6B and 7C), as

mutations in their SH3 domains removed both proteins from the neck (Fig 7C) (Hallett et al, 2002). Nba1 is a potential docking site for Boi1/2 as it binds to both SH3 domains and arrives at the neck at roughly the same time as Bem1 (Fig 3) (Meitinger et al, 2014). Accordingly, a deletion of *NBA1* or of its Boi1/2-binding site (Nba1$_{PxxPΔ}$) removed Boi2-GFP completely, and 55% of Boi1-GFP (Fig 7B). The Nba1-mediated interaction between Gps1 and Boi1/2 suggests that Gps1 anchors the Nba1–Boi1/2–Bem1–Cdc24 complex at the neck (Fig 3F and E). Accordingly, *gps1Δ* cells lacked Nba1-GFP at the bud neck and reduced neck localizations of Boi1-GFP and Boi2-GFP to a similar extent as *nba1Δ* cells (Fig 7A and B) (Meitinger et al, 2014). The localization of the isolated SH3$_{Boi1}$ or SH3$_{Boi2}$ mirrored the SH3 dependencies of the full-length proteins (Fig 7B). However, $t_{1/2}$ of FRAP of SH3$_{Boi1}$-GFP was significantly shorter than $t_{1/2}$ of the full-length Boi1-GFP ($t_{1/2}$ = 0.7 s versus $t_{1/2}$ = 12 s; Fig 7D), indicating that regions beyond SH3$_{Boi1}$ contribute to neck localization. Bem1 was reported to directly bind to Nba1 (Meitinger et al, 2014). However, Boi1$_{PxxPΔ}$-GFP lacking the binding site to Bem1 still displayed a $t_{1/2}$ of 10 s at the bud neck that is very similar to the $t_{1/2}$ of FRAP of the wild-type protein (Fig 7D). We conclude that Boi1 recruitment to the neck is distinct from its synergistic recruitment to the tip.

Bem1-GFP and Cdc24-GFP completely disappeared from the bud neck of cells expressing SH3 mutations in both Boi proteins (*boi1$_{WK}$ boi2$_{WK}$*, Fig 7C). In contrast, *nba1$_{PxxPΔ}$*- or *gps1Δ* cells still kept ~69% of Cdc24-GFP and 58% of Bem1-GFP at the neck (Fig 7C). Whereas Boi2 was solely attached by Nba1-Gps1, 45% of Boi1 were still visible in *nba1$_{PxxPΔ}$*-, or *gps1Δ* cells.

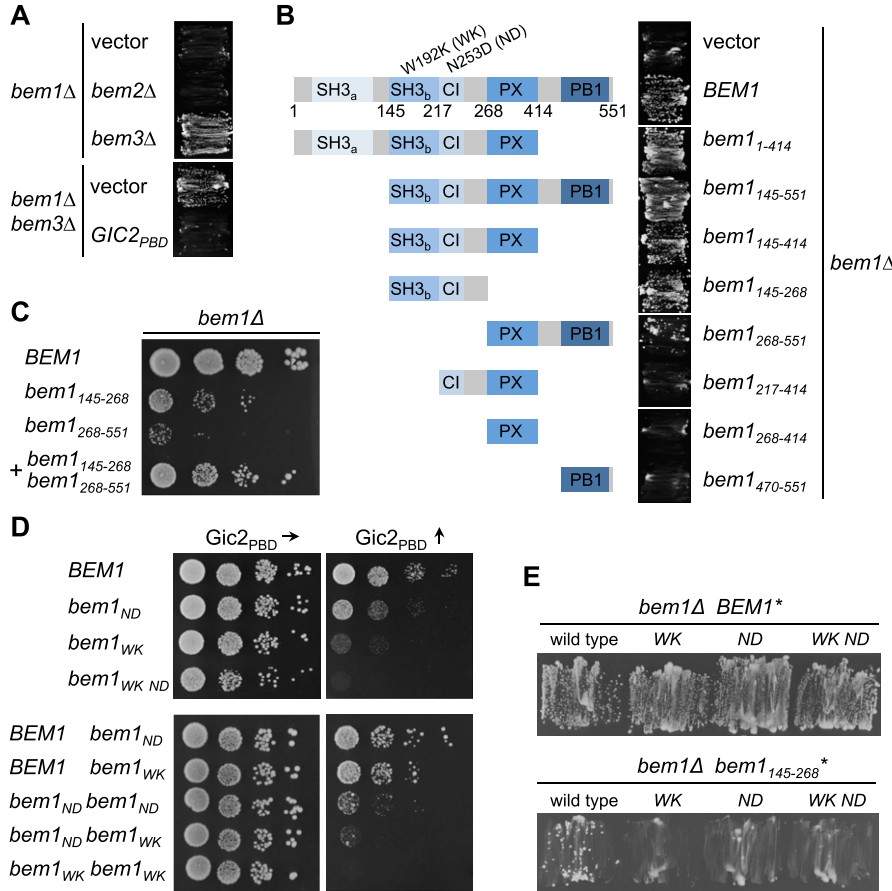

**Figure 4. Bem1 contains two functionally independent regions.**
**(A)** *bem1Δ* cells expressing a vector-encoded *BEM1* and carrying an additional gene deletion as well as an empty vector, or a vector expressing Gic2$_{PBD}$, were incubated on media selecting against the presence of the plasmid-encoded *BEM1*. **(B)** *bem1Δ* cells carrying a vector-encoded *BEM1* and a vector-expressing Bem1 or the indicated fragments of *BEM1* (left panel) were incubated on medium selecting against the vector-encoded *BEM1* (right panel). **(C)** *bem1Δ* cells expressing *BEM1* or the indicated fragments of *BEM1* were spotted in 10-fold serial dilutions onto the SD medium selecting for the presence of the plasmids and incubated at 37°C. **(D)** Haploid cells (upper panel), or diploid cells (lower panel) carrying the indicated alleles of *BEM1* were incubated in 10-fold serial dilutions on media inducing the expression of Gic2$_{PBD}$ to moderate (left panels) or high (right panels) levels. **(E)** *bem1Δ* cells carrying a vector-encoded *BEM1* and additionally expressing the full-length *BEM1* with the indicated residue exchanges (upper panel), or fragments of *BEM1* with the indicated residue exchanges (lower panel), were incubated on medium selecting against the vector-encoded *BEM1*.

Which protein is the alternative anchor for Boi1 at the site of cell separation? The neck localization as well as its interaction with Boi1/2 makes Fir1 a candidate for this role (Brace et al, 2019; Tonikian et al, 2009). Split-Ub analysis confirmed the complex between Boi1/2 and Fir1 and could further demonstrate that the interactions depend on the functional SH3$_{Boi1}$ or SH3$_{Boi2}$ and the predicted SH3$_{Boi1/2}$-binding motif in Fir1 (Fig 7E) (Tonikian et al, 2009). We next introduced the GFP fusions of Boi1, SH3$_{Boi1}$, and the Cdc24–Bem1 complex in strains lacking *FIR1* (*fir1Δ*), lacking the Boi1-binding motif in *FIR1* (*fir1$_{PxxPΔ}$*), or in cells lacking the motifs in *FIR1* and *NBA1* (*nba1$_{PxxPΔ}$ fir1$_{PxxPΔ}$*) (Fig 7B and C). Quantifying the intensities of the GFP signals proved that Fir1 recruits the Boi1–Bem1–Cdc24 complex independently of Nba1 to the bud neck. Boi1 was equally distributed between Nba1 and Fir1 (Fig 7B). In contrast, proportionally more of Cdc24 was anchored through Fir1 than through Nba1, whereas the amount of neck-anchored Bem1 did not change upon removal of the Boi1-binding site in Fir1 (Fig 7C and F for a model).

### Temporal dissection of the Bem1 interaction network

To find out whether Bem1 might bring together different interaction partners at different phases of the cell cycle, we characterized the time dependency of a subset of its interactions through Split-Ub analysis using two spectrally different fluorescent proteins as sensors for interaction (SPLIFF) (Moreno et al, 2013). Here, the C$_{ub}$ is sandwiched between the auto-fluorescent mCherry and GFP (CCG) (Moreno et al, 2013). Upon interaction-induced reassociation with an N$_{ub}$ fusion, the GFP is cleaved off and rapidly degraded. The subsequent local increase in the ratio of red to green fluorescence indicates where and when the direct or indirect interaction between both proteins took place (Moreno et al, 2013).

A Bem1-mCherry-Cub-RGFP fusion protein (Bem1CCG) was expressed from its genomic locus under the control of the conditional *MET17* promoter in *MATa* cells. All N$_{ub}$ fusions were expressed in α-cells from their native promoters, except N$_{ub}$-Cdc42 and N$_{ub}$-Exo70 that were under control of the non-induced P$_{CUP1}$ promoter. N$_{ub}$-Rsr1 was included in the analysis as it generated under its native expression levels a strong interaction signal with Bem1CRU (Fig S4F). Mating and fusion of the a- and α cells allowed Bem1CCG and the respective N$_{ub}$ fusion to interact. Green and red fluorescence were then measured during one cell cycle at the site of cell fusion (PCDI), at the cell front during bud site assembly and bud growth (PCDII), and finally at the bud neck from completion of acto-myosin ring contraction until cell abscission (PCDIII) (Fig S4G). The ratios of the fluorescence intensities (IFs) from individual single-cell experiments were fitted into a single curve and plotted as percentage of N$_{ub}$-induced conversion of Bem1CCG to Bem1CC against time after cell fusion (Fig 8 and Table S1). A regression-based significant positive slope over two time intervals was taken

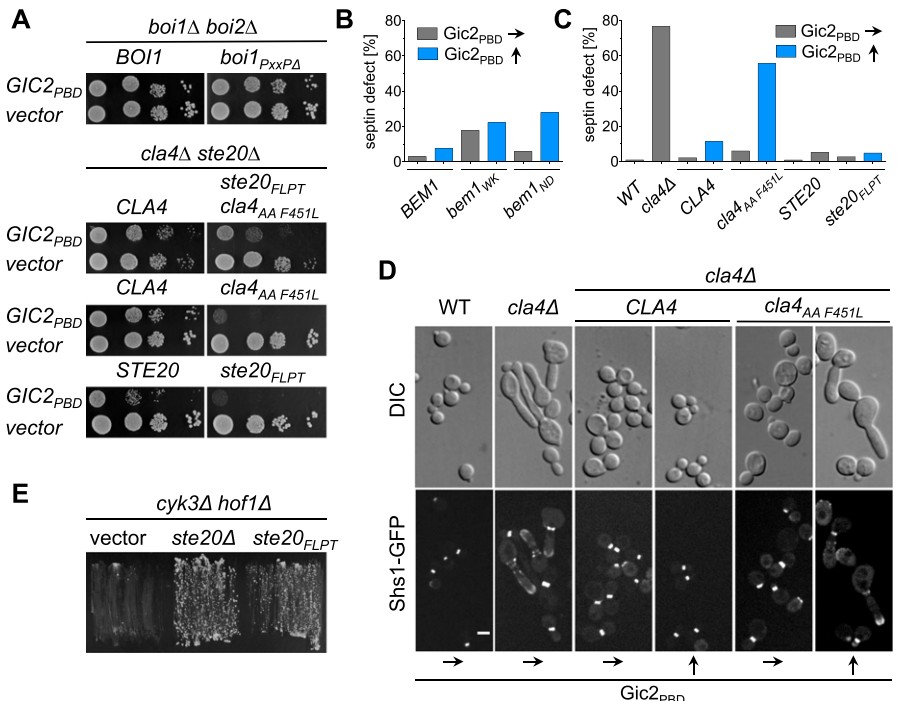

**Figure 5.   The connection between Bem1 and the PAKs is essential upon overexpression of Gic2$_{PBD}$.**
**(A)** Yeast cells carrying the indicated alleles and either Gic2$_{PBD}$ or an empty vector were spotted in 10-fold serial dilutions on medium inducing high levels of Gic2$_{PBD}$. **(B)** Cells expressing Shs1-GFP, the indicated alleles of *BEM1*, and Gic2$_{PBD}$ were incubated under conditions of low (70 $\mu$M Met, gray bars) or high expression levels (0 Met, blue bars) of Gic2$_{PBD}$. Cells (500 < n < 600) were classified according to their native-like or abnormal distribution of the Shs1-GFP. **(C)** As in (B) but with cells (500 < n < 600) carrying the indicated alleles of *STE20* or *CLA4*. **(D)** Microscopy of the cells of (C). Upper panel: DIC channel. Lower panel: GFP channel. Scale bar indicates 3 $\mu$M. **(E)** *cyk3*Δ *hof1*Δ cells expressing *HOF1* from an extra-chromosomal vector and carrying the indicated alleles of *STE20* or an empty vector were incubated on media selecting against the *HOF1*-containing vector.

as evidence for interaction (Figs 8B and S5 and Table S2). No significant increase or a decrease in the relative amount of conversion was considered as absence of interaction. It is important to remind that the absence of an interaction signal indicates "no interaction" only within the detection limit of SPLIFF. The interaction partners of Bem1 fall into the following categories: Ste20, Cdc24, Rsr1, Boi1, Boi2, and Cdc42 interacted with Bem1 during all three phases (Figs 8A and B and S5 and Tables S1 and S2). Cla4 interacted with Bem1 only during PCDI and II. Bud6 interacted with Bem1 only during bud formation and growth (PCDII). Nba1 interacted with Bem1 shortly during PCDI and throughout cytokinesis (PCDIII), whereas Exo70 interacted during PCDII, and in PCDIII, only shortly before cell separation (Figs 8A and B and S5 and Table S2). The multimerization of Bem1 could be observed during a single time frame in PCDI and throughout PCDII. We can, further, differentiate between Bem1 interactions that last through the entire PCDII (Bem1, Boi1, Exo70, Cdc42, and Cdc24) and those that are detectable in small buds only (Bud6 and Rsr1) (Figs 8A and B and S5 and Table S2). The interaction between Bem1 and Rga2 stalled during bud formation and picked up after 10 min into bud growth to continue as long as Bem1 remained at the cortex (Figs 8 and S5 and Tables S1 and S2). The interaction signals between Bem1 and Boi1, Boi2, Cla4, and Ste20 reached a plateau after ~20 min into bud growth. The slight increase of conversion was considered as sign of a continuing yet diminished interaction between Bem1 and Boi1 during the remaining phase of bud growth (Figs 8A and B and S5 and Table S2). The decrease in the ratio of converted Bem1CCG in the N$_{ub}$-Ste20–, N$_{ub}$-Cla4–, and N$_{ub}$-Boi2–expressing cells might already indicate a loss of interaction between Bem1 and the N$_{ub}$ fusions during the transition from bud growth to mitosis. However, it has to be noted that conversion ratios at or above 80% are very difficult to interpret and that no increase or a slight decrease do not necessarily have to

reflect the absence of interaction. During cytokinesis, Boi1/2, Ste20, Rsr1, Cdc24, Cdc42, Exo70, and Nba1 were detectably associated with Bem1 (Figs 8 and S5 and Tables S1 and S2), whereas N$_{ub}$-Cla4, N$_{ub}$-Bud6, N$_{ub}$-Rga2, and N$_{ub}$-Bem1 did not convert Bem1CCG during abscission (PCDIII, Figs 8A and B and S5 and Tables S1 and S2).

## SH3$_{Boi1}$ switches between interaction partners during the cell cycle

The detection of the Bem1–Bem1 interaction requires the oligomerization of the Boi proteins, whereas the proximity between Bem1 and Nba1 is mediated by the simultaneous interactions of both proteins with Boi1/2 (Figs 2 and 3).

We tested the consistency of our SPLIFF analysis by measuring Boi1CCG against N$_{ub}$-Boi1 and N$_{ub}$-Nba1. In agreement with the time dependency of Bem1 multimerization and the formation of the Nba1–Bem1 complex, Boi1CCG was converted to Boi1CC by N$_{ub}$-Boi1 during bud growth and not during abscission, whereas Boi1CCG was converted by N$_{ub}$-Nba1 only during abscission (Figs 8B and 9C and Tables S1 and S2). Boi1CCG/N$_{ub}$-Nba1, thus, lacked the interaction signal observed between Bem1CCG and N$_{ub}$-Nba1 during PCDI. The reported Boi1/2–independent interaction between Bem1 and Nba1 might account for this apparent discrepancy (Fig 8) (Meitinger et al, 2014). The high level of Boi1CCG conversion at the beginning of PCDIII might indicate that Boi1/2-Bem1 arrive together with Nba1 as a preformed complex at the bud neck. The time resolution of our assay cannot distinguish this scenario from our preferred interpretation that conversion occurred exclusively at the neck.

Besides binding to Nba1 and Fir1, SH3$_{Boi1/2}$ interact with additional proteins and might expand the influence of the Bem1–Cdc24 complex to further processes (Tonikian et al, 2009; Kustermann

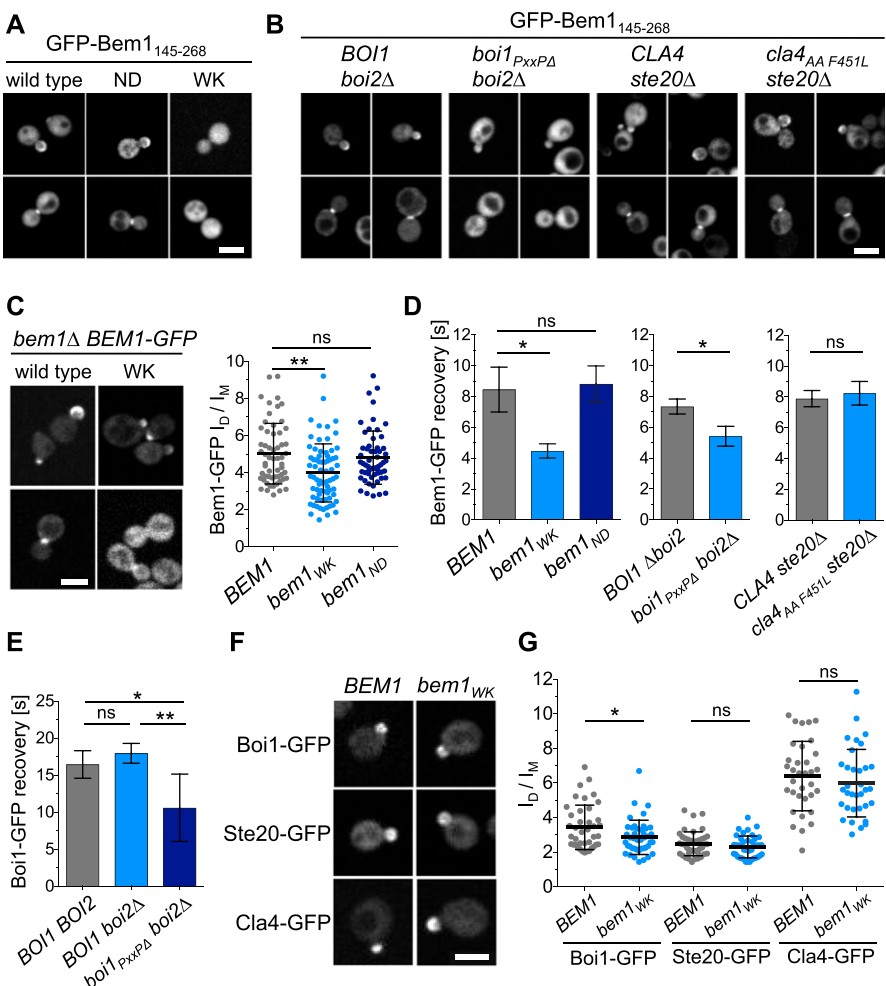

**Figure 6. Boi1 and Boi2 localize Bem1-Cdc24 at bud tip and neck.**
**(A)** Wild-type cells expressing GFP-Bem1$_{145-268}$ (left panel) or GFP-Bem1$_{145-268}$ carrying the ND (middle panel) or WK (right panel) exchange were inspected by fluorescence microscopy. **(B)** Cells of the indicated genotypes expressing GFP-Bem1$_{145-268}$ were inspected by fluorescence microscopy (left panel). Only *boi2Δ boi1$_{PxxPΔ}$* cells show a clear misdistribution of GFP-Bem1$_{145-268}$. **(C)** Left panel: *bem1Δ* cells expressing Bem1-GFP or Bem1$_{WK}$-GFP were inspected by fluorescence microscopy. Right panel: Quantification of the intensity ratios of Bem1-GFP (n = 58), Bem1$_{WK}$-GFP (n = 72), and Bem1$_{ND}$-GFP (n = 57) in bud and mother cells. **(D)** Half-times of fluorescence recovery after photo-bleaching the bud of cells expressing Bem1-GFP or its mutants. Left panel: Bem1-GFP (n = 11), Bem1$_{WK}$-GFP (n = 18), or Bem1$_{ND}$-GFP (n = 24). Middle panel: Bem1-GFP in *boi2Δ BOI1* cells (n = 20) or *boi2Δ boi1$_{ΔPxxP}$* cells (n = 16). Right panel: Bem1-GFP in *ste20Δ CLA4* cells (n = 16) or *ste20Δ cla4$_{PPAAFL}$* cells (n = 14). **(E)** As in (D) but with *BOI1 BOI2 cells* (n = 23), *boi2Δ BOI1* cells (n = 24), or *boi2Δ boi1$_{PxxPΔ}$* cells (n = 22) expressing GFP fusions to *BOI1* or *boi1$_{PxxPΔ}$*. **(F)** Bem1 cells (left panel), or Bem1$_{WK}$ cells (right panel) expressing GFP fusions to Boi1 (upper row), Ste20 (middle row), or Cla4 (lower row), were inspected by fluorescence microscopy. **(F, G)** The ratios of the fluorescence intensities of bud and mother cells from (F) were quantified in *BEM1-* and *bem1$_{WK}$* cells expressing Boi1-GFP (n = 40, 43), Ste20-GFP (n = 43, 40), or Cla4-GFP (n = 34, 34). ns, not significant. * = *P* < 0.05, ** = *P* < 0.01. Scale bars indicate 3 *μM*.

et al, 2017). To test whether other SH3$_{Boi}$ interactions are also cell cycle–specific, we investigated the interaction between Epo1 and Boi1. Epo1 links the cortical endoplasmic reticulum to the polarisome and was shown to bind Boi1/2 (Neller et al, 2015). Mutations that inactivate SH3$_{Boi1}$ or a mutation of the predicted Boi1-binding motif in Epo1 abolished the interaction between both proteins (Fig 9A). A pull down of this binding motif with a GST fusion to SH3$_{Boi1}$ proved its direct interaction (Fig 9B). Epo1 and Nba1 thus compete for the same binding site in Boi1. SPLIFF picked up the interaction between N$_{ub}$-Boi1 and Epo1CCG for the first time during bud growth (Figs 9C and S5 and Tables S1 and S2). No interaction could be recorded for Epo1 during PCDIII at the bud neck (Fig 9C). Although sharing the same interaction site, Epo1 and Nba1 interacted with Boi1–Bem1 at different stages of the cell cycle.

## Discussion

Bem1 is a central scaffold protein for the Cdc42 pathway that is essential in some but not all *Saccharomyces cerevisiae* strains (Dowell et al, 2010). Eliminating the GAP activity of Bem3 and thus

increasing the concentration of active Cdc42 at the cortex especially during bud formation rescues the otherwise lethal deletion of *BEM1* in the strain JD47, whereas the overexpression of a Cdc42$_{GTP}$- and membrane-binding fragment of Gic2 counteracts the positive effect of the *BEM3* deletion (Knaus et al, 2007). Whether yeast cells of a certain strain can live without Bem1 thus seems to depend on the remaining concentration of active Cdc42 at the cortex. A central fragment of Bem1 harboring the SH3$_b$ domain with its neighboring Cdc42$_{GTP}$ binding element and a C-terminal fragment, containing the PB1 and PX domain, independently rescue a *bem1Δ* strain. The C-terminal fragment binds strongly to Cdc24 but does not connect Cdc24 to Cdc42 effector proteins or to the cortex. We propose, in line with published data, that the C-terminal fragment increases the concentration of Cdc42$_{GTP}$ by stimulating the activity of Cdc24 (Shimada et al, 2004; Smith et al, 2013; Rapali et al, 2017). The central SH3$_b$CI fragment interacts with active Cdc42 and four Cdc42 effectors and requires both of its binding sites to rescue *bem1Δ* cells. Our genetic analysis and published data suggest that the functional units SH3$_b$, CI, and PB1 of Bem1 co-operate and form a chain of binding sites that funnel active Cdc42 from its source to its targets (Kozubowski et al, 2008).

Ste20 and Cla4 are the essential ligands of SH3$_b$CI under limiting concentrations of Cdc42$_{GTP}$ (Fig 5A). How can the isolated SH3bCI

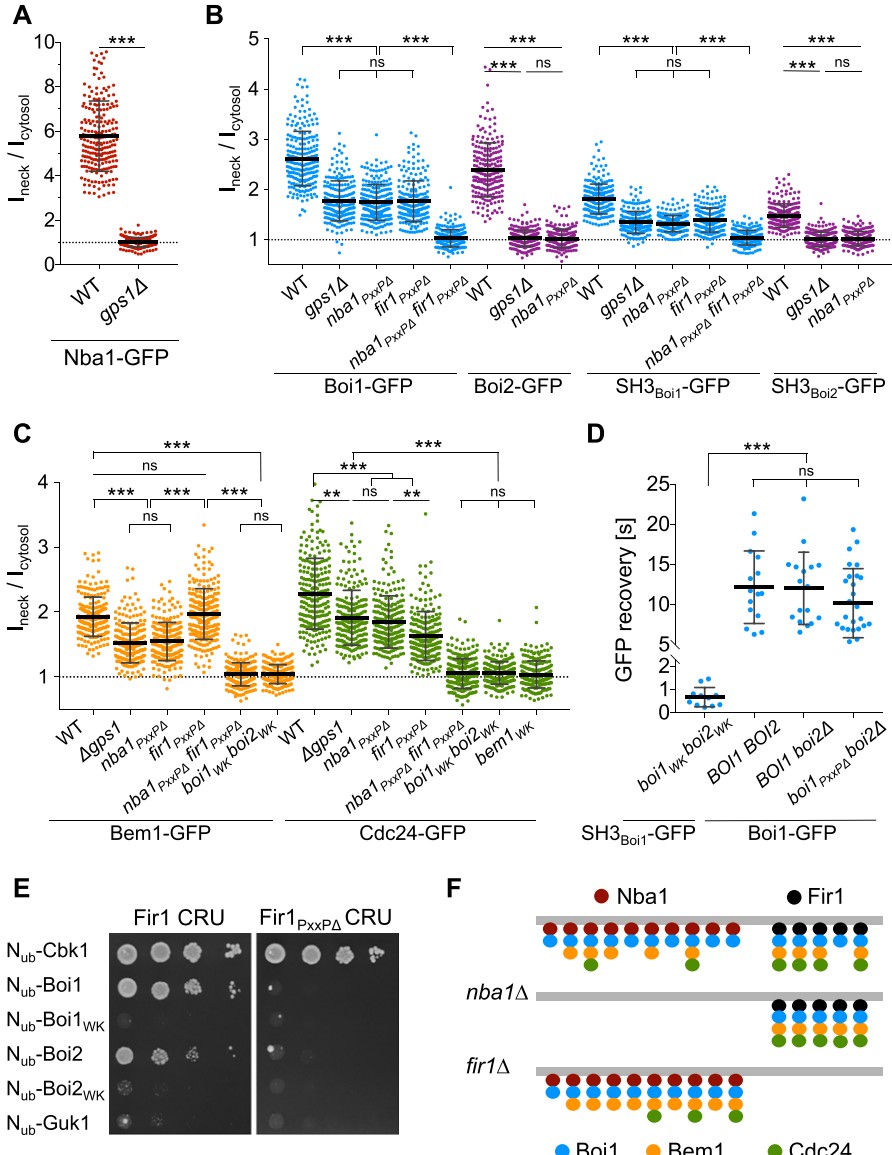

**Figure 7. Two receptor systems attach Bem1-Cdc24 to the bud neck.**
**(A)** Ratios of bud neck to cytosolic fluorescence intensities of Nba1-GFP in wild type and $gps1\Delta$ cells. **(B)** As in (A) but in cells of the indicated genotypes expressing Boi1-GFP, Boi2-GFP, $SH3_{Boi1}$-GFP, and $SH3_{Boi2}$-GFP. **(C)** As in (A) but in cells of the indicated genotypes expressing Bem1-GFP or Cdc24-GFP. **(D)** Fluorescence recovery after photo bleaching the bud neck of $boi1_{WK}$ $boi2_{WK}$ cells expressing $SH3_{Boi1}$-GFP (n = 11), of wild type cells expressing Boi1-GFP (n = 15), of $boi2\Delta$ cells expressing Boi1-GFP (n = 17), of $boi2\Delta$ cells expressing $Boi1_{PxxP\Delta}$-GFP (n = 27). **(E)** Split-Ub analysis as in Fig 2A but with cells expressing Fir1CRU or $Fir1_{PxxP\Delta}$CRU together with indicated $N_{ub}$ fusions. $N_{ub}$-Cbk1: positive control; $N_{ub}$-Guk1: negative control. **(F)** Anchoring the Bem1–Cdc24 complex at the bud neck in wild type- (upper panel), $nba1\Delta$- (middle panel), and $fir1\Delta$ cells (lower panel). Nba1 (red dots) and Fir1 (black dots) recruit Boi1 (blue dots) (for simplicity only Boi1 is shown), which recruits the Bem1–Cdc24 complex (orange and green dots). To explain the impact of the mutations on the distributions of the different proteins, we assume that Nba1 outnumbers Fir1 at the neck, that Nba1 and Fir1 are saturated by Boi1/2, the number of Nba1 and Bem1 molecules are similar, and that Nba1 reduces the affinity between Cdc24 and Bem1 (Meitinger et al, 2014).

without connection to Cdc24 still stimulate the PAKs? A comparison between the binding characteristics of the non-essential Boi1/2-PxxP sites and the essential Ste20/Cla4-PxxP sites suggests a molecular mechanism. Bem1 and all its SH3b ligands are concentrated at the cell tip during bud formation and growth. The cortex localizations of the PAKs but not of Boi1/2 strictly depend on $Cdc42_{GTP}$ (Peter et al, 1996; Leberer et al, 1997; Wild et al, 2004; Kustermann et al, 2017). The isolated binding motifs of Cla4 and of Boi1/2 bind with similar affinities to SH3b, whereas the SH3b-binding motif of Ste20 displays a significantly higher in vitro affinity (Gorelik & Davidson, 2012). SPLIFF analysis shows that both PAKs and Boi1/2 interact with Bem1 during bud formation and growth. Nevertheless, and in contrast to Boi1/2, both PAKs do not measurably contribute to the cortical localization of $SH3_b$CI or of full-length Bem1. We hypothesize that Bem1 interacts stronger with the inactive PAKs than with their cortex-localized $Cdc42_{GTP}$-bound

forms. To explain the stimulatory activity of $SH3_b$CI, we postulate that $SH3_b$CI might open the CRIB domains of the PAKs to actively load them with the CI-attached $Cdc42_{GTP}$ (Lamson et al, 2002). The $Cdc42_{GTP}$-bound CRIB domain might then release the auto-inhibition of the kinases and at the same time, impair the interaction with $SH3_b$. This sequence describes Bem1 not as a passive scaffold but more similar to the kinase scaffold Ste5 as a coactivator that regulates through binding the activity of the PAKs and stimulates the synthesis and the transfer of Cdc42 (Bhattacharyya et al, 2006). Support for our model comes from two observations: 1. Bem1 needs the fully functional SH3bCI to activate Ste20 and Cla4 also during osmostress (Chang et al, 1999; Tanaka et al, 2014). 2. Scd2, the Bem1 homolog from *Schizosaccharomyces pombe,* binds with its second SH3 domain the Ste20 homologs Shk1 and thereby increases the auto-phosphorylation activity of the kinase (Chang et al, 1999; Tanaka et al, 2014).

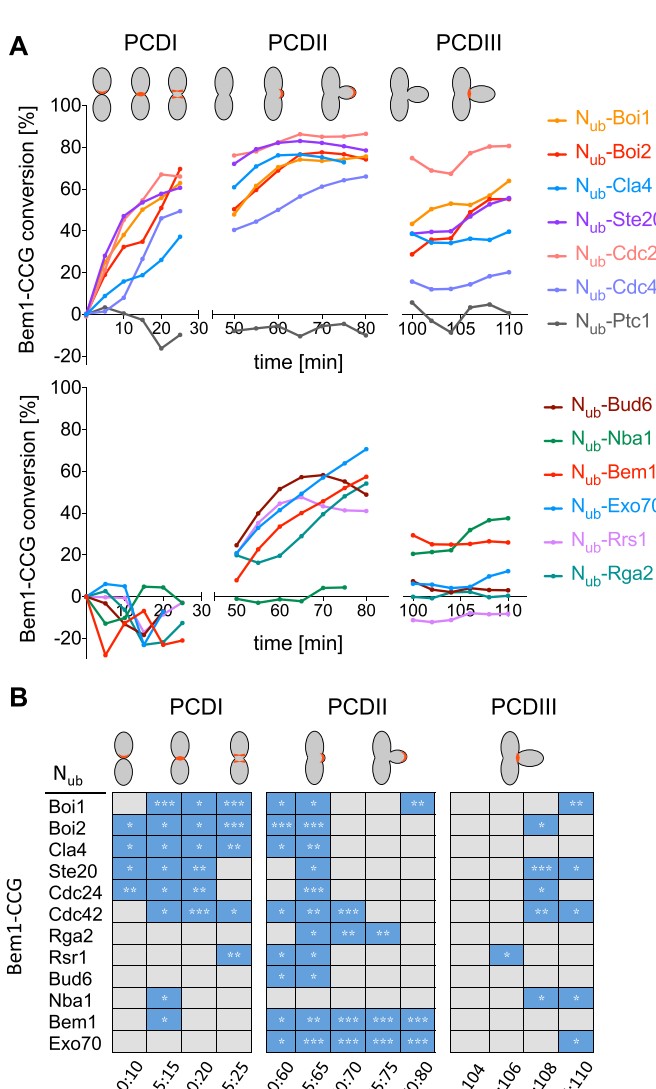

**Figure 8. SPLIFF analysis of yeast zygotes formed by the fusion of a-cells expressing Bem1CCG and α-cells expressing the indicated $N_{ub}$-fusions.**
**(A)** Plotted are the conversions of Bem1CCG to Bem1CC (%) over time. GFP- and mCherry fluorescence intensities were measured at sites of polarized Bem1 locations as indicated in red in the cartoons of the upper panel. Shown are fitted lines calculated from single-cell experiments. **(B)** Interaction profiles for Bem1. Blue boxes indicate a significant increase in conversion over the indicated time. For values below 70% of conversion, significant slopes have to be 1 (% conversion/min) or larger. For values above 70%, slopes have to be positive and significant. To be counted as interaction, the criteria must not be met by the negative control in the respective time window (*** = $P$-value < 0.001; ** = $P$-value < 0.01; * = $P$-value < 0.05).

Localizing Cdc24, stimulating its activity, and channeling Cdc42 to its effectors are separable activities of Bem1 that cooperate in the full-length protein to generate and read the gradient of active Cdc42 (Smith et al, 2013; Woods et al, 2015; Rapali et al, 2017). A temporal map of the interaction network of this complex might thus reveal where at a given time the activated Cdc42 is preferentially directed. SPLIFF is one of the very few techniques that allow analysis of temporal aspects of protein–protein interactions in living cells. The cleavage of GFP from the mCherry-$C_{ub}$-GFP coupled

interaction partner gives a robust ratiometric output for interaction. However, to define the time point of interaction, one has to record the change of this value over time in single cells. The closer the chosen time points, the smaller the change and the less sensitive the measurement. To prove the significance of the observation, measurements have to be repeated in different cells at comparable positions of the cell cycle. In addition, the depletion of the uncleaved mCherry-$C_{ub}$-GFP fusion upon interaction reduces the maximal response during the later time points of the measurements. These features limit the sensitivity of the assay and sometimes blur the distinction between no or rarely occurring interactions. Despite these shortcomings, the application of SPLIFF provided a unique temporal interaction profile of the scaffold protein Bem1. As Bem1 links active Cdc42 to its effectors, Fig 10 summarizes the cellular flow of Cdc42$_{GTP}$ through the cell cycle. During bud site formation and bud growth, Cdc42 is channeled directly to Exo70 and possibly from Boi1/2 to the other Cdc42-activated exocyst component Sec3. Boi1/2 were shown to recruit Bud6 and Bni1 to sites of active exocytosis (Glomb et al, 2020). The temporal formation of the Bem1–Boi1/2–Bud6 complex might thus boost the Bni1-catalyzed actin filament formation and organization during bud site assembly and in small buds (Adams et al, 1990; Dong et al, 2003; Glomb et al, 2020). As the binding sites of Bem1 for Boi1/2 and Exo70 do not overlap, a supercomplex that stimulates and coordinates actin assembly and vesicle fusion during bud assembly and early growth seems plausible (Adamo et al, 2001; Liu & Novick, 2014; Kustermann et al, 2017; Glomb et al, 2020). This complex disassembles in large buds and does not detectably form during cytokinesis (Figs 8 and 10).

The PAKs Ste20 and Cla4 contact Bem1 at the same site as Boi1/2 and form alternative, exclusive interaction states (Fig 10). The Cla4-Bem1-Cdc42-Cdc24 and Ste20-Bem1-Cdc42-Cdc24 interaction states coexist with the Boi1/2-Bem1-Cdc42-Cdc24 interaction states except during abscission where only Ste20-Bem1-Cdc42-Cdc24 and Boi1/Boi2-Bem1-Cdc42-Cdc24 are detectable (Figs 8 and 10). The persistent activation of Ste20 correlates with its role during cytokinesis and its early, CDK-independent activation in the next cell cycle (Moran et al, 2019).

The Boi1/2–Bem1–Cdc24 complex is anchored by Nba1 and additionally by Fir1 to the bud neck (Fig 7F). Fir1 is known to delay cell separation by inhibiting the cell separation kinase Cbk1 (Brace et al, 2019). *NBA1* is synthetic lethal to a certain allele of the essential cytokinesis factor IQGAP (in yeast: *IQG1*), suggesting that Nba1 stimulates abscission (Tian et al, 2014). A better understanding of the functions of Nba1 and Fir1 might reveal how the Bem1/Cdc24 complex moderates cytokinesis.

The PAKs perform many other functions besides their roles during cytokinesis and bud formation (Drogen et al, 2000; Hofken & Schiebel, 2002; Tanaka et al, 2014). The general Cdc42 sensitivity of cells lacking the Bem1-binding sites in *STE20* and *CLA4* indicates that the PAK-Bem1-Cdc42-Cdc24 interaction states are the operative units for many if not all PAK activities.

# Materials and Methods

### Growth conditions and cultivation of yeast strains

All yeast strains were derivatives of JD47, a descendant from a cross of the strains YPH500 and BBY45 (Sikorski & Hieter, 1989; Bartel et al,

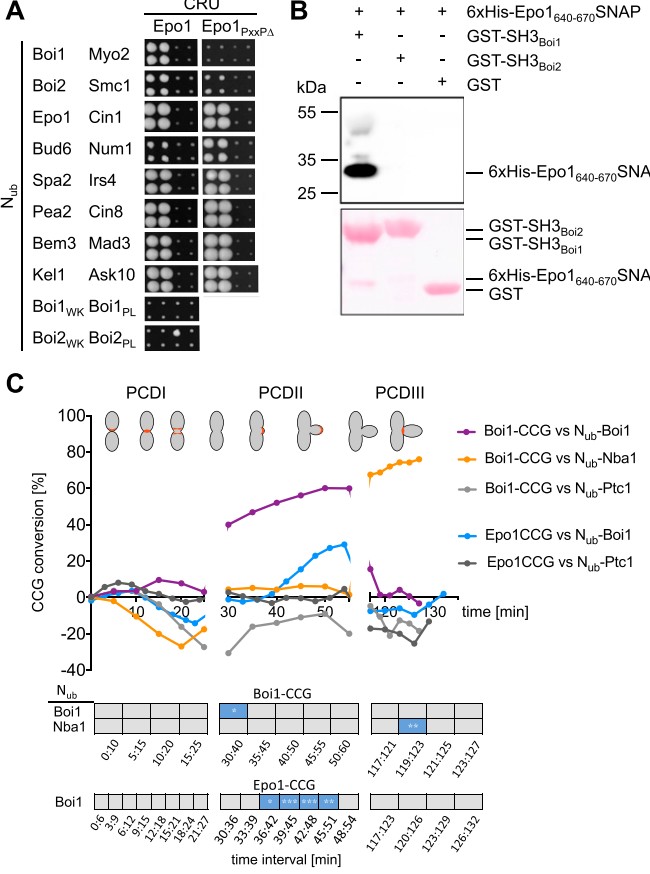

**Figure 9. Epo1 interacts with Boi1 during budding.**
**(A)** As in Fig 1A but with cells expressing Epo1CRU or Epo1$_{654-661\Delta}$ (Epo1$_{PxxP\Delta}$) CRU with the indicated N$_{ub}$ fusions. **(B)** Extracts containing a 6xHIS SNAP-tag fusion to Epo1$_{640-670}$ were incubated with GST-Boi1-, GST-Boi2-, or GST-coupled beads. Glutathione-eluates were stained with Ponceau (lower panel), and anti-His antibodies (upper panel) after SDS–PAGE and transfer onto nitrocellulose. **(C)** SPLIFF analysis: a-cells expressing Epo1CCG or Boi1CCG were mated with α-cells expressing the indicated N$_{ub}$ fusions. Upper panel: The measured fluorescence intensities were processed and visualized as in Fig 8A. Significance of slopes of fitted lines across time intervals are shown in the lower panel (*** = *P*-value < 0.001; ** = *P*-value < 0.01; * = *P*-value < 0.05).

1990; Madura et al, 1993; Dohmen et al, 1995). Cultivation of yeast was performed in standard SD or YPD media at 30°C or the indicated temperatures as described (Dünkler et al, 2012). Media for split-ubiquitin interaction assay and selection for the loss of centromeric *URA3*-containing plasmids comprised 1 mg/ml 5-fluoro-orotic acid (5-FOA; Formedium).

## Construction of plasmids, gene fusions, and manipulations

Construction of N$_{ub}$ and C$_{ub}$ gene fusions as well as GFP-, mCherry-, or mCherry-C$_{ub}$-RGFP (CCG) fusions was performed as described (Wittke et al, 1999; Dünkler et al, 2012; Moreno et al, 2013; Neller et al, 2015). Bem1CRU/-GFP/-CCG were constructed by genomic in-frame insertions of the *GFP*-, *CRU*-, or *CCG* modules behind the coding sequences of *BEM1* or its alleles. In brief, a PCR fragment of the C-terminal region of the respective target gene lacking the stop codon was cloned via *Eag*I and *Sal*I restriction sites in front of the CRU-, GFP-, mCherry-, or CCG-module

on a pRS303, pRS304, or pRS306 vector (Sikorski & Hieter, 1989). Plasmids were linearized using unique restriction sites within this sequence and transformed into yeast cells for integration into the genomic target ORF. Colony PCR with diagnostic primer combinations was used to verify the successful genomic integration. Centromeric plasmids expressing different fragments of *BEM1* were obtained by ligation of PCR fragments spanning the respective region of *BEM1* behind the sequence of the *P$_{MET17}$*-GFP module on the pRS313 vector (Table S3) (Sikorski & Hieter, 1989). Mutations in the coding region of *BEM1*, *STE20*, *CLA4*, or *BOI1* were obtained by overlap-extension PCR using plasmids containing the corresponding ORFs as templates.

Insertion of mutations into the *BEM1*, *STE20*, *BOI1*, or *CLA4* loci were performed in yeast strains lacking the ORFs of the respective genes but still containing their 5′ and 3′ UTR sequences. Mutations were introduced in the respective genes on an integrative pRS vector containing the up- and downstream sequences of the gene (Sikorski & Hieter, 1989). Yeast strains lacking the corresponding ORF were then transformed with the mutated gene on the integrative vector linearized in the promoter sequence of the gene. Successful integration was verified by diagnostic PCR.

Alternatively, insertion of genomic mutations was achieved by CRISPR/Cas9 (Laughery et al, 2015). To introduce the mutations at the chosen sites, guideRNA sequences were cloned into pML plasmids and co-transformed with oligonucleotides harboring the desired mutations. Successful manipulations were verified by PCR product sequencing of the respective genomic ORFs. Details of the introduced mutations are listed in Table S4.

In certain strains, the native promoter sequence was replaced by *P$_{MET17}$* through recombination with a PCR fragment generated from pYM-N35 and primers containing sequences identical to the respective genomic locations at their 5′ ends (Janke et al, 2004). GST fusions were obtained by placing the ORF of the respective gene or gene fragment in frame behind the *Escherichia coli GST* sequence on the pGEX-2T plasmid (GE Healthcare) using *Bam*HI and *Eco*RI restriction sites. Fusions to the human O6-Alkyl-DNA transferase (SNAP-tag; New England Biolabs) were expressed from plasmid pAGT-Xpress, a pET-15b derivative (Schneider et al, 2013). Gene fragments were inserted in frame into a multi-cloning site located between the upstream *6xHIS*-tag–coding sequence and the downstream *SNAP*-tag–coding sequence. The *6xHIS*-tag fusions were obtained by placing the ORF of the respective gene or gene fragment behind the *E. coli 6xHIS*-tag sequence on the previously constructed pAC plasmid (Schneider et al, 2013).

Gene deletions were performed by one step PCR-based homologous recombination using pFA6a natNT2, pFA6a hphNT1, pFA6a kanMX6, pFA6a CmLEU2, and pFA6a HISMX6 as templates (Bähler et al, 1998; Longtine et al, 1998; Janke et al, 2004; Schaub et al, 2006). Lists of plasmids and yeast strains used in this study can be found in Tables S3 and S4 of the supplemental information. Plasmid maps can be obtained upon request.

## Split-Ub interaction analysis

Bem1 was fused to the C-terminal half of Ubiquitin followed by Ura3 carrying an arginine at its N terminus to create Bem1-C$_{ub}$-RUra3 (Bem1CRU). Upon co-expression of a binding partner of Bem1 carrying the N-terminal half of Ub (N$_{ub}$-X) at its N terminus, N$_{ub}$ and C$_{ub}$ are brought into close proximity and the native-like Ubiquitin is reconstituted from its two halves. Ub-specific proteases cleave off the

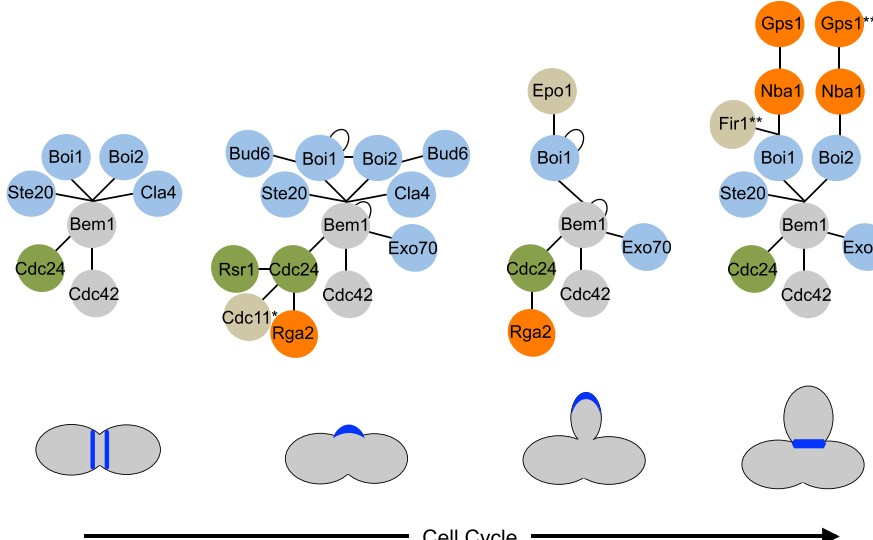

**Figure 10. Interaction networks of Bem1 during PCDI (10–20 min), early and late PCDII (50–55 min, 60–80 min), and PCDIII (104–110 min).** Green color indicates proteins that promote, orange color indicates proteins that reduce active Cdc42. Blue color indicates effectors of Cdc42 or proteins that bind to effectors (Bud6-Bni1). Epo1 binds to the same site of Boi1 as Nba1 or Fir1 but at a different cell cycle phase. *The time point of interaction between Bem1-Cdc24-Cdc11 was obtained from a previous study (Chollet et al, 2020). **The time point of Fir1- and Gps1 binding to Bem1 were indirectly derived through their effects on Bem1 localization (Fig 7).

RUra3 module from $C_{ub}$ and the exposed arginine initiates the degradation of RUra3. As a consequence, cells have no Ura3 activity and do not convert 5-FOA in the toxic 5-fluoro-uracil. The growth of cells on media containing 5-FOA is thus taken as evidence of interaction between the $C_{ub}$- and $N_{ub}$-coupled fusion protein (Johnsson & Varshavsky, 1994; Wittke et al, 1999).

### Array analysis

A library of 548 different α-strains each expressing a different $N_{ub}$ fusion were mated with a *BEM1-$C_{ub}$-R-URA3* (Bem1CRU), *BEM1$_{WK}$–$C_{ub}$–R-URA3* (Bem1$_{WK}$CRU), or *BEM1$_{PB1Δ}$-$C_{ub}$-R-URA3* (Bem1$_{PB1Δ}$CRU) expressing a-strain. Diploids were transferred as independent quadruplets on SD media containing 1 mg/ml 5-FOA. Expressions of the $N_{ub}$ fusions were under control of the copper inducible $P_{CUP1}$-promoter. Media contained different concentrations of copper to adjust the amount of the $N_{ub}$ fusions (Dünkler et al, 2012).

### Individual Split-Ub interaction analysis

CRU and $N_{ub}$ expressing strains were mated or co-expressed in haploid cells and spotted onto the medium containing 1 mg/ml FOA and different concentrations of copper in four 10-fold serial dilutions starting from OD$_{600}$ = 1. Growth at 30°C was recorded every day for 3–5 d.

### Complementation analysis

*bem1Δ* cells expressing *BEM1* from a *URA3*-containing centromeric vector and fragments of *BEM1* from an *HIS3*-containing vector were either streaked directly or spotted as 10-fold serial dilutions onto SD His⁻ media containing 1 mg/ml 5-FOA. As 5-FOA is converted by Ura3 to the toxic 5-fluoro-uracil, the medium counter-selects against the presence of the *BEM1*-expressing *URA3* vector.

### Mating efficiency

Saturated cultures of JD47 cells containing the respective allele of *STE20* and expressing Gic2$_{PBD}$ from a centromeric plasmid under the control of a $P_{MET17}$ promoter and JD53 cells carrying a Kanamycin-tagged *PTC1* gene and expressing Gic2$_{PBD}$ were resuspended in media containing no or 70 µm methionine and grown for 6 h at 30°C. Cells were adjusted to an OD$_{600}$ of 1 and equal amounts of JD53 and JD47 cells mixed and incubated for 4 h at 30°C. Cells were diluted 1/20 and 250 µl of each mating were spread on media selecting for diploids. Colony numbers were counted after 2 d at 30°C.

### Preparation of yeast cell extracts

Exponentially grown yeast cell cultures were pelleted and resuspended in yeast extraction buffer (50 mM Hepes, 150 mM NaCl, and 1 mM EDTA) with 1× protease inhibitor cocktail (Roche Diagnostics). Cells were lysed by vortexing them together with glass beads (threefold amount of glass beads and extraction buffer to pellet weight) 12 times for 1 min interrupted by short incubations on ice. The obtained yeast cell extracts were clarified by centrifugation at 16,000*g* for 20 min at 4°C.

### Recombinant protein expression and purification from *E. coli*

All proteins were expressed in *E. coli* BL21DE3 cells. GST-Bud6$_{1-320}$ was expressed at 30°C for 5 h in LB medium after induction with 1 mM IPTG. GST fusions to SH3 domains of Boi1 and Boi2 and 6xHis-Nba1$_{202-289}$-SNAP were expressed at 18°C in the SB medium for 20 h after induction with 0.1 mM IPTG. Cells were pelleted, washed once with PBS, and stored at −80°C until lysis. All subsequent purifications were carried out on an Äkta Purifier chromatography device (GE Healthcare). Cells expressing GST fusion proteins were resuspended in PBS containing protease inhibitor cocktail (Roche Diagnostics) and lysed by lysozyme treatment (1 mg/ml, 30 min on ice), followed by sonication with a Bandelin Sonopuls HD 2070 (Reichmann Industries service). Extracts were clarified by centrifugation at 40,000*g* for 10 min at 4°C, and the proteins were purified using a 5-ml GSTrap column (GE Healthcare) and subsequent size exclusion chromatography on a Superdex 200 16/60 column versus

HBSEP buffer (10 mM Hepes, 150 mM NaCl, 3 mM EDTA, and 0.05% Tween 20, pH 7.4). Purified protein was concentrated and stored on ice.

6xHis-Nba1$_{202-289}$-SNAP–expressing cells were lysed in IMAC buffer (50 mM KH$_2$PO$_4$, 300 mM NaCl, and 20 mM imidazole containing protease inhibitor cocktail) as described above, and enriched protein was obtained by imidazole gradient elution from a 5-ml HisTrap HP column (GE Healthcare), followed by size-exclusion chromatography.

### GST pull-down assay

GST-tagged proteins were immobilized on Glutathione Sepharose beads (GE Healthcare) directly from *E. coli* extracts. After 1-h incubation at 4°C with either yeast extracts or purified proteins under rotation at 4°C, the beads were washed three times with the respective buffer. Bound material was eluted with GST elution buffer (50 mM Tris and 20 mM reduced glutathione) and analyzed by SDS–PAGE followed by Coomassie staining and immunoblotting with anti-His (dilution: 1:5,000; Sigma Aldrich), or anti–GFP-antibodies (dilution 1:1,000; Roche Diagnostics).

### Surface plasmon resonance measurements

Binding affinities were measured using purified and immobilized GST-SH3$_{Boi1}$ or GST-SH3$_{Boi2}$ as ligands on an anti-GST chip on a Biacore X100 device (GE Healthcare). HBSEP buffer was used as the running buffer in all experiments. The chip was prepared by covalent coupling of an anti-GST antibody (GE Healthcare) as capture molecule to the dextran surface of both flow cells of a CM5 chip (GE Healthcare) using an amine coupling kit (GE Healthcare). GST-tagged ligand proteins were captured on the detection flow cell of the chip, and free GST was captured on the reference flow cell. Purified 6xHis-NBA1$_{202-289}$-SNAP as an analyte molecule was prepared in suitable concentrations in running buffer and kinetics were measured with constant flow over the previously prepared chip. Regeneration after each cycle was achieved by a 20-s injection pulse with 10 mM glycine, pH 2.0. The equilibrium binding constant K$_D$ was subsequently determined by the X100 evaluation software using background subtracted sensograms. All measurements were performed at least as triplicate.

### Fluorescence microscopy

For microscopic inspection, yeast cells were grown overnight in SD medium, diluted 1:8 in 3–4 ml fresh SD medium, and grown for 3–6 h at 30°C to mid-log phase. About 1 ml culture was spun down, and the cell pellet resuspended in 20–50-$\mu$l residual medium. 3 $\mu$l was spotted onto a microscope slide, and the cells were immobilized with a coverslip and inspected under the microscope. For time-resolved imaging, 3 $\mu$l of prepared cell suspension was mounted on an SD-agarose pad (1.7% agarose), embedded in a customized glass slide, and sealed by a coverslip fixed by parafilm stripes. Imaging was started after 15–30 min recovery at 30°C. SPLIFF and other time-lapse experiments were observed under a wide-field fluorescence microscope system (DeltaVision; GE Healthcare) provided with a Olympus IX71 microscope, a steady-state heating chamber, a CoolSNAP HQ2 and CascadeII512-CCD camera both by Photometrics, a U Plan S Apochromat 100 Å ~ 1.4 NA oil ∞/0.17/FN26.5 objective and a Photofluor LM-75 halogen lamp. Images were visualized using softWoRx software (GE Healthcare) and adapted z series at 30°C.

Exposure time was adapted to the intensity of GFP and mCherry signal for every fluorescently labeled protein to reduce bleaching and phototoxicity. Further analyses used an Axio Observer spinning-disk confocal microscope (Zeiss), equipped with an Evolve512 EMCCD camera (Photometrics), a Plan-Apochromat 63 Å~/1.4 oil DIC objective, and 488- and 561-nm diode lasers (Zeiss). Images were analyzed with the ZEN2 software (Zeiss).

### Quantitative analysis of microscopy data and SPLIFF measurements

Microscopy data were processed and analyzed using ImageJ64 1.49 software. For standard fluorescence signal quantification, three regions of interest (ROIs) were determined, first the signal of interest (e.g., tip, bud neck), second a region in the cytosol, and third a randomly chosen position outside of the cell (background). The mean gray values of each ROI ($I_{fluorescence}$, $I_{cytosol}$, and $I_{background}$) were quantified after z-projection. To compare the fluorescence signals of a protein in certain strains, the relative fluorescence ($I_{relative}$) signal of the protein was calculated after subtraction of the background.

$$I_{relative} = I_{fluorescence} - I_{background} / I_{cytosol} - I_{background}$$

### SPLIFF analysis

SPLIFF analysis for temporal and spatial characteristics of Bem1-, Boi1-, and Epo1-CCG interactions was performed as described (Moreno et al, 2013; Dunkler et al, 2015). a-cells expressing the P$_{MET17}$ promoter–controlled CCG fusions were grown in SD medium without methionine and mixed with *MAT α* cells expressing N$_{ub}$-HA fusions either under their native or the P$_{CUP1}$-promoter. After mixing, the cells were immobilized on an SD agarose pad and mating-induced interaction was monitored by three channel z-stack (5 × 0.6 $\mu$m) microscopy every 2, 3, or 5 min. z-slices with fluorescence signals were projected by SUM projection. The FI of mCherry and GFP channels were determined by integrated density measurements of the ROI and a region within the cytosol. For each time point and channel, the intracellular background was subtracted from the localized signal to obtain the localized fluorescence intensity (FI$_{red}$ and FI$_{green}$). The values were normalized to the time point before cell fusion. The resulting relative fluorescence intensity RFI(t) was then used to calculate the conversion FD(t):

$$FD(t) = RFI_{red} - RFI_{green} / RFI_{red}$$

FD(t) as a readout of CCG- to CC conversion describes its temporal progress in percent. Excel was used for initial calculations.

### Regression and slope estimation of SPLIFF analysis

To test significance of increment of percent conversion over time, the non-parametric local regression (loess) (R Core Team, 2019) was first fitted by taking all biological replicates across the entire time window. The fitted line was then used for the generalized additive model (Hastie, 2019). For the generalized additive model, two time intervals with one time point sliding window were used to estimate the slope of rate of change of percent of conversion. Positive slopes with a *P*-value cutoff of 0.05 were considered as statistically significant.

### FRAP experiments

FRAP experiments were performed as described elsewhere with an iMIC digital microscope with a 60× objective (Till Photonics) at RT (Phair et al, 2004). Pictures were acquired with a series of five z-slices each separated by 0.5 $\mu$M. Four images were taken before the ROI was bleached with 100% laser power, a dwell time of 1.2 s/$\mu m^2$, a line overlap of 42%, and an experimental loop count of 10–20. Pictures were taken at a constant time interval of 0.9 s after bleaching except SH3$_{Boi1}$-GFP where signal recovery was measured each 0.25 s in a single z-layer. Initial z-slice projection and fluorescence quantification was performed with the software iMIC Offline analysis. Alternatively, an Axio Observer spinning-disc confocal microscope (Zeiss), equipped with a Zeiss Plan-Apochromat 63 Å~/1.4 oil DIC objective, a 488-nm diode lasers, and an UGA-42 photo-manipulation system was used (Rapp OptoElectronic). Initial signal measurements were ImageJ based. Subsequently, all data sets were double-normalized using Excel. The software Prism 6.0 (GraphPad) was used for the fitting of the double-normalized data to a one-phase association curve.

### Statistical evaluation

GraphPad Prism was applied for statistical data evaluation. The distributions of the data sets were analyzed by the D'Agostino and Pearson normality test. t tests were used to analyze data following a normal distribution, whereas Mann–Whitney U tests were used for data that did not pass these criteria. The one-way ANOVA or the Kruskal–Wallis ANOVA tests were used to compare data sets from more than two groups.

## Supplementary Information

## Acknowledgements

We thank S Timmermanns and N Schmidt for technical assistance. The work was funded by grants from the Deutsche Forschungsgemeinschaft to N Johnsson (Jo 187/5-2,-3; Jo 187/9-1).

### Author Contributions

S Grinhagens: conceptualization, investigation, and visualization.
A Dünkler: conceptualization, supervision, investigation, visualization, and methodology.
Y Wu: conceptualization and investigation.
L Rieger: investigation.
P Brenner: investigation.
T Gronemeyer: investigation.
MA Mulaw: formal analysis and methodology.
N Johnsson: conceptualization, supervision, funding acquisition, investigation, and writing—original draft.

### Conflict of Interest Statement

The authors declare that they have no conflict of interest.

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
