## [Reviewer comments · Life Science Alliance]

Life Science Alliance

A time resolved interaction analysis of Bem1 reconstructs the flow of Cdc42 during polar growth

Soeren Grinhagens, Alexander Dünkler, Yehui Wu, Lucia Rieger, Philipp Brenner, Thomas Gronemeyer, medhanie mulaw, and Nils Johnsson

DOI: <https://doi.org/10.26508/lsa.202000813>

Corresponding author(s): Nils Johnsson, Ulm University

Review Timeline:

Submission Date:	2020-06-12
Editorial Decision:	2020-07-10
Revision Received:	2020-07-15
Editorial Decision:	2020-07-17
Revision Received:	2020-07-20
Accepted:	2020-07-21

Transaction Report:

Please note that the manuscript was previously reviewed at another journal and the reports were taken into account in the decision-making process at Life Science Alliance.

Reviewer #1 Review

Comments to the Authors (Required):

This manuscript entitled, "A time resolved interaction analysis of Bem1 reconstructs the flow of Cdc42 during polar growth" describes a series of protein-protein interaction assays focusing on the scaffold protein Bem1, a protein that plays a well characterized role in cell polarization by forming a complex with the GEF for Cdc42, Cdc24 and a Cdc42 binding kinase, Cla4 or Cdc20. The authors use split Ub-based interaction assays to map the interaction between a number of polarity proteins and derivatives thereof. The nature of these interaction assays are not resolved in either time or space. Notwithstanding this major issue, the interaction data are quite detailed and in some cases make the effort to distinguish direct and indirect interactions. This information confirms a lot of results in the field and extends them somewhat. In general the manuscript is not clearly organized and dense; there are parts that are poorly written. Overall, if properly written, this manuscript would be a solid contribution to the field, but it would be a better fit for a journal with a systems biology focus than this journal.

1 - The authors claim to have mapped Bem1 interaction partners in time and space. This central claim "we systematically mapped its protein interactions in time and space" is not supported by the data. The only data that provides time resolved information concerning protein-protein interaction is the SPLIFF analyses in figures 8 and 9. The time resolution of these data is generally ~5 minutes. However, Bem1 exchanges very rapidly on the order of seconds (PMID 15353546), thus there is a fundamental mismatch between the dynamics of the assay and the processes under study.

1b - No raw image data from the SPLIFF analyses is shown, rather 10 pages of tables of processed numbers are shown.

2 - The authors do not appropriately contextualize their work in light of the published literature. "Although being intensely studied the molecular mechanisms behind Bem1's effect on polarization as well its precisely regulated cellular distributions are not fully understood (Atkins et al., 2008; Kozubowski et al., 2008; Li and Wedlich-Soldner, 2009)." This statement indicates there has been no significant insight into "Bem1's effect on polarization as well its precisely regulated cellular distributions" in the last decade, which is incorrect. Including the very relevant finding that the essential function of Bem1 can be provided by fusing Cdc24 to Cla4 (PMID 19013066, though this paper is cited, its central conclusion is not mentioned).

"Accordingly, changes in the composition and structure of their interaction network should correlate with the different phases of cell growth. This assumption was not yet tested....Bem1 might also provide a positive feedback for polarity establishment and might serve as a platform for a negative feedback during later stages of the cell cycle"

There are papers in the field that address the cell cycle regulation of this complex (PMID 11113154) and other work extend this and demonstrates that Bem1 directly induces positive feedback specifically at Start (PMID 28682236).

3 - There appears to be some ambiguity concerning the strain background. This is relevant because the viability of Bem1 deficient strains depends upon strain backgrounds. In some backgrounds, Bem1 Δ mutants are very sick but viable (PMID 1538785), in W303 backgrounds they barely sporulate (PMID 26426479), in combination with *rsr1* Δ they are dead (PMID 14625559).

The MS states

"All yeast strains were derivatives of JD47, a descendant from a cross of the strains YPH500 and BBY45 (Dohmen et al., 1995)"

Following the citations:

Dohmen et al., 1995 states, "All other *S. cerevisiae* strains were derivatives of JD47-13C (MAT a leu2-3, 112 lys2-801 his3- Δ 200 trp1- Δ 63 ura3-52) (Madura et al., 1993)"

Madura et al., 1993 lists JD47-13C as Mat a, ura3, trp1, leu2 with the source "this work" yet its relationship to YPH500 and BBY45 is not mentioned.

4 - In figure 4, the growth of the PX and PB domain fragment (Bem1268-551) appears far weaker than the controls and the 145-268 fragment. The growth of these strains should be compared at least using serial dilution spot assays.

Minor points:

In figure 1, some strains were apparently plated in a different order than the majority of strains. The authors should flip those panels to conform to the logical order (and the control should be labeled).

The writing is not clear and there are numerous poorly phrased sentences.

P.4 "The interaction is highly affine and can be reconstituted with bacterially expressed proteins in vitro" presumably the authors mean this is a direct high affinity interaction.

P. 3 "The specific functions of Bem1 are controversially discussed."

p. 7 Bud6 nucleates together with the yeast formin Bni1 linear actin filaments

P. 8 Consistently, Nub-Nba1 Δ PxxP also failed to interact with the CRU fusion of Bem1 but not with the CRU fusion of Nba1 (Fig. 3B, C).

Consistently is used incorrectly.

P.9 Δ bem1-cells can be rescued by the simultaneous (??) deletion of the Cdc42 GAP ... The ability to rescue depends on Bem3's GAP activity and a functional PH domain (Fig. S1).

Mutation of Bem3 GAP domain or impairing its interaction with lipids is sufficient to restore viability to Bem1 Δ cells.

p. 10 "A low concentration of active Cdc42 requires the presence of Bem1 for cell survival (Fig. 4A). We artificially reduced the pool of free Cdc42GTP by expressing increasing amounts of Gic2PBD in bem1WK-, bem1KA-, bem1ND- or bem1WK ND-cells (Fig. 4D)"

While over expression of Gic2PBD may reduce the pool of free Cdc42GTP, the authors do not provide direct evidence that this is the case, other alternatives are possible. It is clear that over expression Gic2PBD compromises cell growth in a variety of strains lacking wt Bem1 function. The authors should be more circumspect in their interpretations and the figures should be labeled accurately (i.e. Gic2 OE instead of Cdc42GTP \downarrow)

P10 "To test whether the closely spaced SH3b and CI operate independently of each other, we introduced the WK and ND mutations in the different BEM1 copies of the diploid genome to co-express Bem1WK and Bem1ND in one cell. The undiminished sensitivity of these cells toward Gic2PBD overexpression suggests that both binding sites operate in cis and have to contact Cdc42GTP and one of the SH3b-ligands at the same time (Fig. 4C)."

If the "binding sites operate in cis and have to contact Cdc42GTP and one of the SH3b-ligands at the same time", then Bem1ND and Bem1WK transheterozygotes would resemble Bem1 Δ /Bem1 Δ which is not the case. The sensitivity to Gic2PBD OE suggests that this is assertion true for Bem1 to function at its most robust.

Reviewer #2 Review

Comments to the Authors (Required):

The manuscript of Grinhagens et al. addresses how the Rho GTPase Cdc42 controls cellular

polarity with temporal and spatial fidelity in budding yeast. The manuscript focuses on the interactions between a Cdc42-associated scaffold protein called Bem1 and effector proteins during the cell cycle. The study primarily employs the use of Split-Ubiquitin assays to tackle the problem, techniques that have been pioneered by the Johnsson lab. The Split-Ubiquitin assays are particularly well suited to the identification of labile interactions between low-abundance proteins that occur in a cell cycle-dependent fashion. These types of interactions have largely eluded detailed analysis by other techniques over the past 25 years despite intensive interest and investigation. As such, this reviewer feels that the study is timely and addresses an important biological question. Moreover, the approaches employed are likely to be of interest to the diverse readership of this journal.

The concept that Bem1 interacts with many proteins is not completely new, as some, although by no means all of the interaction partners described in the current study have previously been identified. What is new is the identification of the mechanisms by which many of these interactions occur, and the use of time-resolved analysis to dissect the temporal regulation of these interactions during the cell cycle. In the opinion of this reviewer, it is primarily the strength of the evidence for the temporal regulation of Bem1 interactions that will determine the impact of the manuscript (see major point 5 below). This point should be strengthened in order to be fully convincing.

Major points:

1. The clarity of the manuscript must be improved by providing the reader with more background information on the methods being used at the beginning of the results section and during the development of the manuscript. While it may be possible for readers to work this out by trawling through multiple citations, surely the authors want their manuscript to be understood by the diverse readership of this journal so that others will adopt their powerful methodology. For example, the authors refer to SPLIFF analysis in the abstract, but the acronym does not appear to be explained in the text. Similarly, in the first line of the results, "We searched for binding partners of Bem1 by performing a systematic Split-Ubiquitin (Split-Ub) screen of Bem1CRU against 548 Nub fusion proteins known or suspected to be involved in different aspects of polarized growth in yeast". I am unsure that Bem1CRU will be comprehensible to many readers. Even after reading the methods section, I still did not understand what the "R" in CRU referred to. Nor did I understand why the cells were counter-selected on 5-FOA or what was meant by the term Bem1CCG. All of this became clearer when I read the Kustermann et al. 2017 and Dünkler et al. 2015 manuscripts, but readers of this journal will not have the time to do this and this manuscript should be comprehensible in its entirety, which is not the case at the moment.

Similarly, the study employs the use of conditional MET17 and CUP1 promoters in different experiments. Please guide readers who are not familiar with budding yeast genetics through the logic of the experiments and their interpretation by stating under which conditions these promoters are either repressed or induced, as had previously been done in the Kustermann et al. 2017 manuscript.

The nomenclature bem1WK and bem1ND should be modified for clarity, since it is not clear from these names in which domains the mutations reside, nor what interactions they perturb. I found myself repeatedly referring back to the text for this information. In order to improve clarity, it would be helpful to include this information in Figure 1B schematically and use nomenclature in the text that is more intuitive, such as bem1SH3bCl.IND and bem1SH3b.WK, or an alternative that the authors feel is appropriate.

2. Figure 3. Sensorgrams for the SPR data presented in Figure 3D (mis-called as 3E in the figure legends) should be provided as supplemental data. It is important to provide the sensorgrams in order to judge how the data points in the graphs in Figure 3D were derived and to judge whether the system reaches saturation. Additionally, the authors mention in the materials and methods that

GST was used as a negative control. Indeed, this controls for non-specific interactions between the analyte and the sensor chip. However, a control demonstrating the specificity of the SH3Boi1 or 2 interaction with 6xHIS-Nba1202-289-SNAP is warranted. The appropriate control would be 6xHIS-nba1 Δ PxxP-SNAP or boi1or2SH3WK versus the wild type partner or the mutant proteins with each other. If the authors opt not to do this control, they should move the SPR data to supplemental data, but the sensorgrams should nevertheless be included.

3. The section of the manuscript titled "Functional annotation of Bem1-PAK interaction states" requires additional editing. Initially, the experiments compare Bem1-Boi complexes with Bem1-PAK complexes, so the title of the section is confusing to the reader. This initial section is very dense and the logic is difficult to follow at the start because the statements did not seem to flow. I would encourage the authors to try and state at the outset the goal of the experiments, then lead the reader through the rationale for their experiments and provide a short summary sentence of what the results appear to indicate in each section. Below are some comments that the authors should follow to try and make this section clearer.

Page 11: " All four SH3b-ligands bind Cdc42GTP." Please specify which four Bem1 SH3b-ligands are being referred to. Please clarify the subjects being referred to in the first four sentences of this paragraph.

Page 11: " The functional linkage between SH3b and Ci suggest that Bem1 shuttles active Cdc42 from Cdc24 directly to these effectors." A functional link could refer to a genetic or physical interaction between the same or different genes/proteins, and the link could be direct or indirect. An alternative sentence, that may be clearer is, "The proximity of the PB1 domain to the juxtaposed SH3b and Ci domains within the Bem1 protein suggests that Bem1 may shuttle active Cdc42 from Cdc24 to its effectors."

Page 11: " The PH domains of Boi1 (PHBoi1) and Boi2 (PHBoi2) bind Cdc42GTP (Bender et al., 1996; Kustermann et al., 2017). This reviewer was unable to find the data demonstrating that the PH domain of Boi2, or full-length Boi2, interacts with Cdc42GTP in the cited work. Please either provide the correct citation or edit this statement and any others that are made relating to this point.

In this section, the authors propose that when active Cdc42 is down-regulated, the connection between Bem1 and PAKs is essential, whereas the connection between Bem1 and the Boi paralogs is not essential. The conclusion derives from the observation that the over-expression of gic2pbd is more toxic in cla4 and/or ste20 poly-proline mutants than over-expression of gic2pbd in Δ boi2 boi1 Δ PxxP mutants in which poly-proline residues are also mutated. This makes two assumptions: first, that the over-expression of gic2pbd is similar in the pak and boi poly-proline mutants. Second, that the pak and boi poly-proline mutants are equally defective in their interaction with the Bem1 SH3b domain. The first point should be controlled using a western blot shortly after over-expression on the gic2pbd construct i.e. before the cells die. The second point is more challenging. In the absence of supporting evidence, this caveat ought to be explicitly acknowledged. In a more general sense, I found myself asking what the physiological significance of these results may be. Under what conditions would cells experience down-regulation of active Cdc42 where the Bem1-PAK interaction becomes critical? Can the authors address this point, or rephrase this section in a manner that will emphasize the biological significance of the results?

Page 12: Figure 5G. In the text there is no conclusion to this section of the manuscript. Is the reader to conclude that the interaction of Boi1 and Bem1 via their respective poly-proline and SH3b domains is critical for bud growth? It does not appear to be. In order to evaluate the contribution of

the Boi1-Bem1 poly-proline - SH3b interaction to bud growth, a control should be provided showing the rate of bud length extension after complete inhibition of Boi1 and Boi2 function using a conditional allele. Also, the Y-axis in graph on the left should be labeled "bud length", not "bud size". The Y-axis in the graph on the right should be "bud length over time", not "bud growth", since bud growth occurs in three dimensions. The appropriate units for bud growth would therefore be μm^3 , not μm .

4. Page 12: "The cortical targeting of SH3bCI requires the ligands of SH3b but not Cdc42GTP, as SH3bCIND-GFP is still concentrated at bud neck and tip, whereas SH3bWKCI-GFP stays cytosolic throughout the cell cycle (Fig. 6A)." The statement assumes that bem1 SH3bCIND is completely deficient in Cdc42GTP binding in vivo. However, the mild phenotype of the bem1 SH3bCIND mutant might suggest that the mutant is not completely deficient in Cdc42GTP binding in vivo (Yamaguchi et al. 2007 Figures 4A and B). Alternatively, if the bem1 SH3bCIND mutant is completely defective in Cdc42GTP binding, this would indicate that the physiological importance of Bem1 in linking Cdc42GTP to effectors is quite subtle. The authors could edit the text in light of these considerations.

5. Page 17: "During cytokinesis only Boi1/2, Ste20, Cdc24, Cdc42 and Nba1 are detectably associated with Bem1 (Fig. 8). Bem1 clearly distinguishes between Ste20 and Cla4 during abscission". The interpretation of these results is critically dependent upon the sensitivity of the SPLIFF assay, which is not possible to judge from these experiments. For example, the authors interpret the results to indicate that Bem1 interacts with Cdc42 during cytokinesis, but not with Cla4 or Exo70. However, the shape of the three curves are very similar, and when the error bars are considered, the differences become even more difficult to interpret. Moreover, looking at the raw data, in only 2 of 8 cells undergoing cytokinesis is an increase in Bem1-CCG conversion observed during 3 or more time points after mating to NUb-Cdc42 cells. Nevertheless, the authors conclude that Cdc42 and Bem1 interact at cytokinesis. The authors may well be accurate in their interpretation, but they would need to demonstrate the sensitivity of their assay before convincing this reviewer that their conclusion is justified. This is a major concern, because the title of the manuscript hinges on this claim. The point could be addressed by performing the SPLIFF assay on two wild type proteins; two mutants displaying a weakened interaction, and finally mutants displaying a strongly perturbed interaction.

Minor points:

Page 3: "Among the many processes that are controlled by Cdc42GTP are the organization of the septin- and actin cytoskeleton, the spatial organization of exocytosis, mating, osmolarity sensing, the mitotic exit, and the regulation of cell separation during cytokinesis (Pruyne et al., 2004; Bi and Park, 2012)." Delete "the" preceding "mitotic exit".

Page 4: "Bem1 consists of two N-terminally located SH3 domains (SH3a, SH3b), a lipid- and membrane-binding PX domain, and a C-terminal PB domain (PBBem1) (Bender et al., 1996; Matsui et al., 1996)." The sentence implies that the PX domain has two separate interactions, one with lipid and one with membrane, which is not the case. Please remove either "a lipid" or "and membrane".

Page 4: "However, the removal of SH3a hardly affects secretory vesicle fusion or other aspects of polarized growth (France et al., 2006)." Could be changed to, "However, the removal of SH3a has little effect on secretory vesicle fusion..."

Page 4: "Although being intensely studied the molecular mechanisms behind Bem1's effect on polarization as well its precisely regulated cellular distributions are not fully understood." Could be

changed to: "Despite being intensely studied, the molecular mechanisms underlying Bem1's effect on polarization and the manner of its regulated cellular distribution are not fully understood..."

Page 6: "The Split-Ub measured interactions between Bem1 and Boi1/2, Nba1 or Bud6 depend on SH3b of Bem1 (Fig. 1A, B)." The sentence should be edited. Also, since the assay is not quantitative, it is not correct to say that it measures an interaction. This should also be changed on page 7 and anywhere else that the term "measured" is used in the context of this assay.

Page 6: " To map the contact sites of the binding partners on the structure of Bem1, we repeated the screen with mutants of Bem1 that either carried the well-characterized W192K exchange in SH3b (Bem1WKCRU) or lacked the C-terminal PB domain and thus the binding site to Cdc24 (Bem1 Δ PBCRU)". The widely accepted nomenclature for referring to mutant proteins in *Saccharomyces cerevisiae* is lower-case i.e. bem1WKCRU and bem1 Δ PBCRU. Please correct this throughout the manuscript.

Page 9: "BEM1 is not essential in each yeast strain but required for cell survival in the strain JD47 (Fig. 4A)". Should be changed to, " BEM1 is not essential in all yeast strains, but is required for cell survival in the strain JD47 (Fig. 4A)".

Page 9: " The CRIB domain of Gic2 (Gic2PBD) captures Cdc42GTP and is used as RFP fusion to monitor active Cdc42 in living yeast cells (Atkins et al., 2013; Brown et al., 1997; Okada et al., 2013; Orlando et al., 2008)." Do the authors intend to say, " The CRIB domain of Gic2 (Gic2PBD) interacts with Cdc42GTP and has been tagged with RFP to monitor active Cdc42 in living yeast cells (Atkins et al., 2013; Brown et al., 1997; Okada et al., 2013; Orlando et al., 2008)"?

Page 13: "Boi1/2 are necessary and sufficient to attach also the full length Bem1 to the bud neck (Fig. 6C, see also Fig. 9) whereas the fluorescence signal of Bem1WK-GFP is only reduced but not abolished at the tip of small and large buds (Fig. 6C)." There is a problem with panel C. Only one of the four sub-panels is labeled WK in my pdf version of the manuscript. It is therefore not possible to assess the statement.

Page 13: "To obtain a quantitative measure of tip affinity we compared the FRAPs between the cortex-localized Bem1-GFP, Bem1WK-GFP, and Bem1ND-GFP (Fig. 6D)." FRAP experiments do not provide a quantitative measure of affinity i.e. Kd. Please tone down the language.

Page 13: "Boi1 and Boi2 are recruited to the bud cortex mainly through their Cdc42GTP- and lipid-binding PH domains." Please provide the supporting citations.

Page 16: "Split-Ub analysis provides a static projection of a measured interaction from all cell cycle stages. To correlate a certain interaction with a cell cycle specific function or localization we had to resolve this static projection into its successive temporal layers." Please rephrase this section as it is unclear.

Page 17: "The interaction signals between Bem1 and Boi1, Boi2, Cla4 and Ste20 reach a plateau after approximately 20 min into bud growth." Should be "Cla4", not "Cal4".

Page 20: "The isolated binding motives of Cla4....". Should be "motifs".

Page 20: "Still and in contrast to Boi1/2 both PAKs do not...". Should be, "Nevertheless, and in

contrast to Boi1/2, both PAKs do not..".

Page 24: "Insertion of mutations into the BEM1, STE20, BOI1, or CLA4 loci were performed in yeast strains lacking the ORFs of the respective gens but still...". Should be "genes".

Page 39: "Shown are the cut outs of the quadruplets expressing the Nub fusion of the interacting protein on the left, next to a fusion that does not interact, except for the cells expressing Nub-Cdc11, -Spa2, and -Exo70, that are shown on the right of cells expressing non-interacting Nub fusions." Do the authors mean Nub-Nba1, -Spa2 and -Exo70? I do not see growth of the Nub-Cdc11 / Bem1 CRU strain.

Page 40: " The cartoon implies an indirect interaction between Bem1 and Gsp1." Please change to Gps1.

Page 47: In Figure 1B, reference to bem1 K482A may be mis-interpreted by readers as referring to a bem1 point mutant that was used to map the interaction with Msb1, Cdc11, Ras1 etc. In fact, a bem1 Δ PB1 mutant lacking the entire PB1 domain was used for the mapping and this should be clarified by removing the " K482A".

Page 48: In Figure 2C, it is not possible to evaluate whether there is any non-specific interaction between GST alone and the GFP-tagged Boi proteins because only the 110 - 200 kDa area of the blot is presented, whereas GST runs at around 27 kDa and would therefore not be present in this part of the blot. It would also be clearer to label the lanes as "bound" and "lysate" or "bound and "unbound" rather than "pellet" and "sn".

Figure 2D: "Model of a potential actin nucleator complex." The protein complex in the model does not include a protein with actin nucleation activity. Either include Bni1 in the model or refer to the complex as a potential regulator of an actin nucleator.

Figure S3: In panel E, I was unable to find a description of what the labels "S" and "E" on the blots refer to.

Throughout the manuscript: The authors use commas for decimalization e.g. 2,35. However, I think that numbers are usually decimalized by a period i.e. 2.35.

Reviewer #3 Review

Comments to the Authors (Required):

This manuscript by Grinhagens et al focus on one important protein in the yeast polarity network, the scaffold protein Bem1. Bem1 is known to localize to sites of polarization and to interact with several binding partners, in particular Cdc42-GTP, its guanine nucleotide exchange factor Cdc24 and Cdc42 effectors, the PAK-family kinases Cla4 and Ste20 and the two Boi proteins Boi1 and Boi2. The paper combines 2-hybrid interaction studies (using the split-ubiquitin system), structure-function analysis, localization and protein dynamics studies (by FRAP analysis) with an assay to measure protein-protein interaction in vivo (called SPLIFF) to define the mode of Bem1 localization at the bud neck. The most interesting outcome is the demonstration that Bem1 interaction partners are specific for distinct cellular locations and thus change over time.

This is an overall solid study, which will be a valuable addition to the field. I am however a bit

puzzled about the main focus on Bem1, when this scaffold was previously shown to be dispensable upon linking Cdc24 with the Cdc42 effector Cla4. There are a number of issues that need to be addressed before the study is suitable for publication. My main comments are detailed below.

A large part of the manuscript describes in detail the interaction landscape of Bem1 with partner proteins identified by a split-ubiquitin screen. This analysis leads them to describe the precise mode of Bem1 anchoring at the bud neck, which exclusively depends on Boi1/2 proteins and not PAK-family kinases. This part of the paper is overall clear, with a few comments for improvements:

- The GST pull-down from yeast extracts shown in fig 2C does not show direct interaction better than the 2-hybrid assay. To test for direct interaction, the authors would need to perform binding experiments with recombinant proteins. They should at least change the text to be more precise.
- In the cartoons shown in Fig 2D and 3F, it would help the reader to introduce the interaction domains involved in the interaction shown. The average reader won't have the domain organization of each of the tested proteins in their head, even if this is described in the text.
- The authors describe localization and dynamic studies of Bem1, showing the importance of Boi1/2, which are themselves anchored by Nba1 and Fir1 to the bud neck. In their Bem1 fragment localization analysis (Fig 6), they show images to illustrate the importance of Boi1/2 for Bem1 bud tip localization. Their analysis leads to the conclusion that Boi1/2 are essential for Bem1 bud neck localization but only part of the Bem1 bud tip localization signal. However, no image is shown of full-length Bem1 in these mutants (perhaps the unlabeled Fig 6C? but I can't understand what this is meant to show). This would be important to add.
- These findings also raise the question of what other determinant contributes to Bem1 localization at the bud tip. Perhaps Cdc24, given the PX-PB fragment displays some function. This could be tested.

The analysis of Bem1 function and the analysis of the minimal functional domain(s) is in my view less satisfying:

- The use of Gic2-PBD as a way to "artificially reduce the pool of free Cdc42-GTP" can be used as a tool to sensitize the cells to reduction in Cdc42-GTP. However, the text should make clear that this is an inference, not a fact. Using the Gic2-PBD probe at lower expression levels and quantifying its intensity at sites of polarization, or using the probe for a Cdc42-GTP pull-down would be better ways.
- More problematic is that this assay, as well as complementation of *bem1Δ* by the various fragments, uses a simple patch growth on a plate, an assay that is not quantitative. The authors claim that the central 145-551 region of Bem1 can be split in two fragments that can each complement. This complementation appears rather a bit weaker. Growth curves, measurements of generation time, position of the bud, cell shape... such quantitative parameters would likely help differentiate and not lump all complementation into one group.
- The experiment in figure 4C co-expressing two different Bem1 alleles in the same diploid cell is over-interpreted. The authors claim that "The undiminished sensitivity of these cells toward Gic2PBD overexpression suggests that both binding sites operate in cis and have to contact Cdc42GTP and one of the SH3b-ligands at the same time". I don't think the assay is quantitative enough (see above) for the authors to distinguish the difference in sensitivity of the two alleles. What they can say is that there is no cross-complementation, but it is difficult to say more about the arrangement of Bem1 molecules in the hypothetical complex.
- The finding that specific mutation that block Bem1 interaction for each of the PAK kinase results in phenotypes that mimic the loss of the corresponding kinase is a nice confirmation of their specific role, but hardly surprising.

The final part of the paper presents a SPLIFF analysis. This is a method, which the authors

previously developed, to map the location of protein-protein interactions in cells. The basic concept is that one of the two binding partners is tagged with a construct consisting of mCherry, a half-ubiquitin and GFP, while the second partner is tagged with the other half of ubiquitin. Upon partner protein binding, ubiquitin refolds and is recognized by the proteasome leading to cleavage and degradation of the C-terminal GFP. Thus, ongoing interaction leads to progressive reduction of the GFP to mCherry ratio (or increase of the opposite ratio). This is a neat assay, which to my knowledge has not yet been used by other research groups. Therefore, it needs a bit more introduction in the text - I had to go read the original publication to understand the method before I could appreciate Figures 8 and 9. Similarly, in the abstract, it would be helpful to describe what the method does (dynamically detect protein-protein interactions in vivo) rather than its acronym, which will be uninformative to most people. The outcome of this analysis is that Bem1 interacts with distinct binding partners at different times and locations. There are a few issues that need to be addressed:

- One important conclusion from the paper is that Bem1 changes binding partners over time. This is derived from the SPLIFF analysis, showing for instance interaction of Bem1 with Exo70 only during bud tip extension, and Bem1 with Nba1 only at the bud neck. An important missing control is whether the relevant proteins (in this example Exo70 and Nba1, but the issue applies also to the other binding partners) are expressed to the same levels at the time when no interaction is detected (i.e. at cytokinesis for Exo70 and during bud growth for Nba1). I also wonder whether one would not get the same result by simply looking at the localization of the binding partners: are those not interacting at the bud tip/neck simply not localizing there?
- The whole SPLIFF analysis lacks statistical analysis. There are error bars provided, but no statistical test. For instance, it is not clear from the graph that the Cla4 curve is flat at PCDIII, but the Cdc42 one isn't. The same comment applies for the claim that some of the interactions do not occur throughout the length of the examined phase. For instance, is the small decrease in the Rga2 signal in the first 5min of PCDII really significant? Finally, the authors indicate interaction between Bem1 and Cdc24 in PCBIII (which is likely), but the curve is mostly flat at 80% level as there is essentially no GFP signal left from the end of PCBII. I am not sure this can be confidently interpreted as meaning interaction at this stage, which the authors also point out, but then leave these more difficult to interpret interactions in their summary panel in Fig 8B. It would be safer to indicate them differently.
- In fig 9, why is the SPLIFF assay performed in different orientation for Boi1-Nba1 (and Boi1-Bem1) than for Boi1-Epo1? It would be better to have the mCherry-C-ub-GFP tag on Boi1 for all assays to allow for cross-comparison.

The discussion could benefit from better structuring, perhaps with sub-headings, and from less speculation. There are a few mistakes:

- It is not clear what data the sentence "We could further show that the CRIB domains of both PAKs are the essential acceptors of Cdc42GTP" refers to.
- The sentence "Sec3. Bud6 binds and stimulates the Cdc42-activated formin Bni1" is inaccurate: Bni1 is activated by various Rho GTPases, but not Cdc42 (see Dong et al, JCB 2003).
- The proposals that "the Gps1-Nba1-Boi1/2-Bem1-Cdc24 complex is an abscission promoting complex, whereas the Fir1-Boi1-Bem1-Cdc24 complex might act as an abscission inhibiting complex" and that Bem1 actively stimulates the transfer of Cdc42-GTP to effector proteins are highly speculative, especially as previous work has shown that the necessity for Bem1 can be alleviated by physically tethering Cdc24 to Cla4 (Kozubowski et al, 2008). Although the Cdc24-Cla4 fusion may not represent the physiological situation, it still indicates that the main (or sole) role of Bem1 is to act as a scaffold. I do not understand why the authors appear to dismiss this evidence, which is also cited in very uncertain terms in the introduction ("Bem1 might also provide a positive feedback").

Minor comments:

Please explain how the 548 prey proteins were selected for the split-ubiquitin screen.

In Fig 1A, presenting all experiments in the same orientation would be helpful so you can label the figure explaining what is being shown: mirror images of those plated the other way should do the trick.

On this figure, why is Cdc42 not interacting with WT full-length Bem1? And is this WT Cdc42 or GTP-locked Cdc42?

In Fig S3, I do not understand what exactly the mutations introduced in Cla4 are: Cla4-F15A AA sounds like a subset of Cla4-F15A AA/PP F451L, but then Cla4-F15A AA does not interact with Nbp2, while Cla4-F15A AA/PP F451L does, so it seems improbable. While I understand well the Ste20 mutations (F470L means F470 replaced by L), the annotation F15AAA/PPF451L is cryptic for me. This needs to be clarified.

Fig 5G lacks error bars. It is also interesting that the bud growth rate is slower for *boi2Δ boi1ΔPxxP* initially, but then appears faster. How is the generation time of these cells?

The images in panel 6C are not described.

There is a mysterious panel 6G, which is not described in either legend or text.

Reviewer #1 Review

Comments to the Authors (Required):

This manuscript represent a thorough and multifaceted analysis of the Bem1 interaction network. The authors have responded appropriately to the thorough and careful comments of all three reviewers. As a result the revised manuscript is greatly improved, though it remains very dense and difficult to digest. In terms of rigor and quality, it meets the standards of this journal, though it is more of an encyclopedic list of interactions as opposed to shedding specific light onto a particular aspect of Bem1 function. Also, there remain issues with the SPLIFF analysis and the lack of clarity is a significant concern. Regardless of the editorial decision, there are a few points that need to be addressed.

1) The authors claim that Bem1 homointeraction is a novel finding. While strictly correct, the close ortholog of Bem1, *S. pombe* Scd2 has been shown to interact previously - indeed the key W residue involved was shown to be important there too (PMID 12409291 also mentioned in 14625559).

Related to Bem1 homooligomerization, the authors state, "The loss of interaction signal between Bem1CRU and Nub-Bem1 in a *boi2Δ boi1ΔPxxP* strain suggests that the Split-Ub detected Bem1-Bem1 interaction is mediated by the multimerization of the Boi-proteins (Figs. 1A, B, 2A) "and subsequently "The detection of the Bem1-Bem1 interaction requires the oligomerization of the Boi-proteins." However, the aforementioned Endo paper shows Bem1 fragments multimerize directly and Fig 2A in the present manuscript reveals that Bem1-N retains the ability to interact, albeit less strongly, in the *boi2Δ, boi1PxxPΔ* background, in contrast to the author's interpretation.

2) Concerning the trans-complementation of WK and ND variants of Bem1. A complementation assay implies that each mutant is phenotypically distinct from wild-type, yet each mutant is able to complement a Bem1 deletion allele very well. The phenotype of these alleles only arise in when Gic2PBD is over expressed, i.e. only in a synthetic background. Given that both the WK and the ND alleles are phenotypically normal in an otherwise wild-type background, it is an over-interpretation to say that "both binding sites *have to operate* within the same Bem1 molecule" and that Bem1 is "a cell cycle specific shuttle that distributes active Cdc42 from its source to its effectors and helps to convert the PAKs Cla4 and Ste20 into their active conformation" as the authors state in the abstract.

3) It is perplexing that in the Spliff assays the CCG conversion appears to change precipitously over time (when comparing PCDII to PCDIII). For example with Nub-Cdc42, at 100 minutes at the bud neck, the % conversion (~20%) is even lower than it was at the site of fusion 70 minutes earlier at 30 minutes (~50%). It is not clear how that could happen unless there is a lot of synthesis of the fusion protein during the experiment. Indeed in figure S5, and table 2 a number of combinations result in a negative slope. What does this mean?

4) The writing is still quite dense, in part due to complex jargon and abbreviations and a number of inartful phrases.

One example of the latter is, "Both mutations did not visibly affect the interactions of Bem1 with Fks1, Cdc42, and Spa2 (Fig. 1A)" which would be more clearly stated as "Neither mutation visibly affected the interaction of Bem1 with Fks1, Cdc42, or Spa2 (Fig. 1A)."

5) it would be worth stating in the first paragraph of the results that the split-Ub assay detects both direct and indirect interactions. This point should be repeated in the context of the Spliff assay.

6) Figure S4D length is misspelled on the y-axis of the first graph.

Reviewer #2 Review

Comments to the Authors (Required):

The authors have made an effort in responding to my comments in an attempt to improve the manuscript. While I am convinced that the study makes a useful contribution to the field, I nevertheless noticed a few points that require additional attention.

I feel that the acronym "SPLIFF" in the abstract should either be defined or removed for clarity. Two reviewers raised this point, yet the authors have opted not to follow their request.

In the Introduction on page 4:

"Instead, Bem1 was also shown to modestly stimulate Cdc24's GEF activity (Woods et al., 2015; Smith et al., 2013; Rapali et al., 2017)." The study by Woods did not demonstrate that that Bem1 stimulated Cdc24 GEF activity. In the context of the sentence, this reference should be removed.

Page 10:

"These phenotypes were recapitulated in bem1WK- or in bem1ND-cells, or in cla4PPAAFL-cells upon overexpression of Gic2PBD (Fig. 5C, D). In contrast, Gic2PBD overexpression did not affect cellular morphology or septin structure of ste20FLPT- or boi2Δ boi1ΔPxxP-cells (Figs. 5C, D, S4C)." The figure call-outs do not seem to be correct. Do the authors mean: "These phenotypes were recapitulated in bem1WK- or in bem1ND-cells, or in cla4PPAAFL-cells upon overexpression of Gic2PBD (Fig. 5B, C, D). In contrast, Gic2PBD overexpression did not affect cellular morphology or septin structure of ste20FLPT- or boi2Δ boi1ΔPxxP-cells (Figs. 5C, S4C)"?

Pages 12 and 13:

Given that 69% of Cdc24-GFP and 58% Bem1-GFP are still found at the neck in gps1Δ and nba1 Δ PXXP cells (Fig. 7C), the section heading "Nba1 anchors Boi1/2-Bem1-Cdc24 at the bud neck" is overstated. This could be modified to, "Nba1 plays a role in tethering Boi1/2-Bem1-Cdc24 to the bud neck."

Page 13:

"In contrast, proportional more of Cdc24", should be "proportionally" more.

Page 14:

"The mCherry-Cub-RGFP module (CCRG) was fused in a-cells behind Bem1 to create Bem1CCG." Could be changed to "A Bem1-mCherry-Cub-RGFP fusion protein (Bem1CCG) was expressed under the control of the conditional MET17 promoter in MATa cells."

Page 14:

"till" should be "until".

Page 14:

"Green and red fluorescence were measured during one cell cycle at the site of cell fusion (PCDI), at the cell front during bud site assembly and bud growth (PCDII), and finally at the bud neck from completion of acto-myosin ring contraction till cell abscission (PCDIII) (Fig. S4E)." Should be Fig. S4G.

Legend to Figure S1. "bem3PHmut harbors the residue exchanges R646S, R645S, K647D that are known to impair the binding of the PH domain to phospholipids." Please provide the reference or edit the statement.

Table S1. As previously pointed out during the first round of review, decimalization should be indicated with a point, not a comma i.e. should be 4.3 and not 4,3. Also, in my version of the manuscript there was yellow highlighting of text in supplementary tables that may not be intended.

Comments to the Authors (Required):

Re-review of Grinhagens et al

I appreciate the authors' effort in addressing my concerns and those of the other reviewers. Unfortunately, I am still uneasy on several aspects of this manuscript, principally the time-resolved analysis of interactions *in vivo*, which forms the crux of the paper. Please find my detailed comments below.

The authors have worked on the text, but it still is difficult to read. First, the logic of the text is still somewhat difficult to follow. It would really help the reader to link thoughts by logical connectors and end paragraphs with one sentence summarizing the take-home message. For instance, the whole description of the SPLIFF results is very difficult to follow. It is just a list of interactions observed at different times, with no effort to explain what it means or characterize them in relation to the previously presented data. The reader is just left to figure it out. Second, the author still fail to provide introductory information where needed: For instance, in the result section, the SPLIFF analysis is simply introduced by saying: "we characterized the time dependency of a subset of its interactions through SPLIFF analysis" and the text proceeds to explain tagging but does not explain straight away the concept of the assay. This is essential.

I appreciate the authors have made an effort in the statistical analysis of the SPLIFF data, but I have to admit that I am not better convinced than I previously was. I remain somewhat unsure of how the analysis is done and whether significance can be derived from individual timepoints. If I understand well the methods, the data for a given cell cycle phase and protein pair is fitted to a regression line whose slope serves to define whether interaction is taking place or not. I do not understand how significance of individual timepoints is obtained, described in the sentence "two time intervals with one time point sliding window were used to estimate the slope of rate of change of percent of conversion". More critically, I am rather convinced that one cannot derive significance at each specific timepoint from the low number of cells imaged (typically less than 10, as shown in table S1) without a clear way to identify cell cycle stages and align the timings of the imaged cells. Just to take a couple of examples: The Bem1-Nba1 is proposed as one interaction that varies at different timepoints: the interaction is negative during budding and is judged to be positive during the last two timepoints of cell division (according to fig 8B, but the text states "throughout cell division"). However, just comparing values and plotting them on a graph show that the data is extremely noisy, with (standard deviation) error bars that would span most of the height of the graph shown in Fig S5. A simple t-test comparing first to last time point reveals no statistical significance. I do not want to imply that the conclusion of the authors is wrong, but I am not convinced from the present dataset.

If one then compare this to the data on Exo70, which is judged to be positive during budding but largely not during division, there is a similar small increase at the end of division, which is in fact is judged positive in Fig 8B. However, the text in the discussion concludes that Bem1-Boi-Bud6-Exo70 super-complex is not detectable during cytokinesis, which is at odds with the data. Third, the Boi1-Nba1 interaction during cell division shown in Fig 9C starts at a very high level of conversion, when this level was very low at the end of budding. This would indicate that interaction already took place before cell division. The controls proposed by reviewer 2 would seem to be necessary to establish a baseline for interpretation of the results. Contrary to what the authors indicate, they have a number of mutants that would be appropriate, for instance Boi1-PxxP (or Boi2-PxxP) mutants should still localize but not interact with Bem1, Bem1-WK can be used at the bud tip as a negative control for interaction with all SH3b-interacting partners, and Bem1-ND can be used as negative control for

the interaction with Cdc42.

I am a little confused by the statement on Bem1 requirement for cell survival in their strain. As pointed out by reviewer 1, *bem1Δ* is lethal in some strains but not in others where it leads to strong sickness. The authors state in the results that *bem1* is required for cell survival in strain JD47 and in the discussion that deletion of *bem1* is lethal, and show a patch of cells that do not grow. How was the strain even obtained if *bem1Δ* is lethal? Similarly, the methods section states that insertion of mutations in *bem1* was performed in strains lacking the ORF. And so it must have been viable! This suggests that *bem1Δ* is not lethal in this strain, and so the description of the phenotype is inaccurate.

Are the experiments in Fig 6A-B that analyze the localization of the Bem1 SH3b-CI fragment performed in *bem1+* or *bem1Δ* background? I suppose it is the former and this should be specified in the text. If in *bem1+*, then the localization likely results (at least in part) from the Boi1/2-induced dimerization with WT Bem1. This would be consistent with the interaction hierarchy described in Fig, 2, but it is misleading to label Boi1/2 as receptors for this fragment without showing direct experimental support. Indeed, examination of the localization of Bem1 full-length protein is not examined in *boi1Δ boi2-PxxP* mutants, which would be necessary. The Bem1-WK localization shown does not directly attribute localization to Boi1/2.

The model proposed by the authors that Bem1 preferentially binds the closed PAKs to promote interaction of Cdc42-GTP, which in turn would release auto-inhibition and block Bem1 binding is interesting, but not directly supported by experiments shown in this manuscript. The only one is that PAKs do not appear to be necessary for the localization of Bem1. The cited data that the *S. pombe* Bem1 homologue promotes PAK auto-phosphorylation is also not entirely consistent with the proposed model, as this data was obtained from immuno-purified complexes, which are not shown to include Cdc42. This data rather suggests that Bem1 binding on its own suffices to release auto-inhibition. In either case, this may only contribute to PAK activation in a minor way as just published data has now shown that in *S. pombe* like in *S. cerevisiae* linking the PAK to the Cdc42 GEF fully bypasses the function of the Bem1-like protein (Lamas et al PLoS Biology 2020).

July 10, 2020

RE: Life Science Alliance Manuscript #LSA-2020-00813-T

Prof. Nils Johnsson
Ulm University
Biology
James Franck Ring N27
Ulm D-89081
Germany

Dear Dr. Johnsson,

Thank you for submitting your revised manuscript entitled "A time resolved interaction analysis of Bem1 reconstructs the flow of Cdc42 during polar growth". We have evaluated the three referee reports from your submission at another journal and your response. We consulted further with Ref 1 - his/her report is attached below.

We are happy in principle to publish this paper in LSA, subject to a revision of the current manuscript to include all the non-experimental/textual issues raised by the referees at the other journal and by Ref 1 now. Please supply a revised manuscript and a short rebuttal that details where and how each substantive point has been addressed in the revised manuscript.

Please also take care of the following formatting requirements:

- please fill in all required information into our system (category, author contributions, summary blurb)
- please make your 'Author Contributions' and 'Conflict of Interest' statements separate sections in your manuscript text
- please add the supplementary figure legends to the main manuscript text and upload your supplementary figures as separate files
- please add a callout for Figure 1C & Figure 7F
- please list 10 authors et al. in your references

A. FINAL FILES:

B. MANUSCRIPT ORGANIZATION AND FORMATTING:

Sincerely,

Reilly Lorenz
Editorial Office Life Science Alliance
Meyerhofstr. 1

69117 Heidelberg, Germany
t +49 6221 8891 414
e contact@life-science-alliance.org
www.life-science-alliance.org

Reviewer #1 (Comments to the Authors (Required)):

This revised manuscript contains publishable results, but it was not a good fit for this journal, due to (i) it required additional revisions beyond the permitted single round for a report (ii) some of the conclusions are somewhat preliminary and (iii) the presentation remained somewhat encyclopedic. However, with inclusion of the proposed revisions in the response to reviewers (this is not the version available from LSA), this manuscript is suited for publication by LSA.

While I was not super impressed with the "Spliff* experiments", they are worth reporting and publishing (while I am not a statistics expert, the explanation the authors provided seems sensible). As the authors indicate, assaying assembly of protein complexes in vivo in a non-perturbing manner (as opposed to colocalization) is difficult and the approach described is an interesting approach to this difficult problem. However, the authors should clarify in the context of their manuscript the limitations of the assay (and the inherent challenges of measuring complex formation in vivo).

* For clarity, the authors should explain the origin of the naming of the "spliff" assay, rather than the reverse engineered naming ("Split-Ub and two spectrally different fluorescent proteins (SPLIFF)"). The acronym makes more sense in the context of reggae music which is strongly associated with red/yellow/green colors and certain smoking habits.

The authors write in the abstract, "The analysis characterizes Bem1 as a cell cycle specific shuttle that distributes active Cdc42 from its source to its effectors and helps to convert the PAKs Cla4 and Ste20 into their active conformation."

This statement suggests a catalytic role for Bem1 is demonstrated. While I find their genetic data convincing that full Bem1 facilitates multiple interactions, there is no direct evidence that this is a catalytic vs a structural role. Bem1 could remain part of the active complexes. The discussion on page 18 provides the relevant context, but this remains hypothetical and should be stated as being a hypothesis in the abstract.

Concerning Rev 1 Point 14

Given that both the WK and the ND alleles are phenotypically normal in an otherwise wild-type background, it is an over-interpretation to say that "both binding sites *have to operate* within the same Bem1 molecule" and that Bem1 is "a cell cycle specific shuttle that distributes active Cdc42 from its source to its effectors and helps to convert the PAKs Cla4 and Ste20 into their active conformation."

Bem1 WK and Bem1 ND alleles and the WK/ND transheterozygotes

- presumably do not support any function that requires both binding sites in the same molecule.
- these alleles are viable in otherwise wild-type yeast cells

At the same time, it is true that, as the authors show, these strains are more sensitive to Gic2 overexpression than the wt molecule, implying that these alleles and the trans-het are somewhat defective. Thus it is fine to suggest that *full* Bem1 function requires both binding sites in the same molecule, but it is not correct to conclude that "Bem1 function *requires* both binding sites in the same molecule" because Bem1 can clearly execute its essential function without these binding sites present in the same molecule.

Response to Reviewer 1

1. "However, the authors should clarify in the context of their manuscript the limitations of the assay (and the inherent challenges of measuring complex formation in vivo)."

Response: We included a paragraph in the discussion section describing the limitations of the SPLIFF assay in the context of our experiments.

2." This statement suggests a catalytic role for Bem1 is demonstrated. While I find their genetic data convincing that full Bem1 facilitates multiple interactions, there is no direct evidence that this is a catalytic vs a structural role. Bem1 could remain part of the active complexes. The discussion on page 18 provides the relevant context, but this remains hypothetical and should be stated as being a hypothesis in the abstract."

Response: We changed the last sentence in the abstract to emphasize the hypothetical character of our statement.

3. "Thus it is fine to suggest that *full* Bem1 function requires both binding sites in the same molecule, but it is not correct to conclude that "Bem1 function *requires* both binding sites in the same molecule" because Bem1 can clearly execute its essential function without these binding sites present in the same molecule".

Response: We agree and changed the conclusion of the experiment accordingly.

July 21, 2020

RE: Life Science Alliance Manuscript #LSA-2020-00813-TRR

Prof. Nils Johnsson
Ulm University
Biology
James Franck Ring N27
Ulm D-89081
Germany

Dear Dr. Johnsson,

Thank you for submitting your Research Article entitled "A time resolved interaction analysis of Bem1 reconstructs the flow of Cdc42 during polar growth". It is a pleasure to let you know that your manuscript is now accepted for publication in Life Science Alliance. Congratulations on this interesting work.

DISTRIBUTION OF MATERIALS:

Again, congratulations on a very nice paper. I hope you found the review process to be constructive and are pleased with how the manuscript was handled editorially. We look forward to future exciting submissions from your lab.

Sincerely,

Reilly Lorenz
Editorial Office Life Science Alliance
Meyerhofstr. 1
69117 Heidelberg, Germany
t +49 6221 8891 414
e contact@life-science-alliance.org
www.life-science-alliance.org